# Society, Materials, and the Environment: The Case of Steel

**Jean-Pierre Birat**

IF Steelman, Moselle, 57280 Semécourt, France; jean-pierre.birat@ifsteelman.eu; Tel.: +333-8751-1117

**Abstract:** This paper reviews the relationship between the production of steel and the environment as it stands today. It deals with raw material issues (availability, scarcity), energy resources, and generation of by-products, i.e., the circular economy, the anthropogenic iron mine, and the energy transition. The paper also deals with emissions to air (dust, Particulate Matter, heavy metals, Persistant Organics Pollutants), water, and soil, i.e., with toxicity, ecotoxicity, epidemiology, and health issues, but also greenhouse gas emissions, i.e., climate change. The loss of biodiversity is also mentioned. All these topics are analyzed with historical hindsight and the present understanding of their physics and chemistry is discussed, stressing areas where knowledge is still lacking. In the face of all these issues, technological solutions were sought to alleviate their effects: many areas are presently satisfactorily handled (the circular economy—a historical' practice in the case of steel, energy conservation, air/water/soil emissions) and in line with present environmental regulations; on the other hand, there are important hanging issues, such as the generation of mine tailings (and tailings dam failures), the emissions of greenhouse gases (the steel industry plans to become carbon-neutral by 2050, at least in the EU), and the emission of fine PM, which WHO correlates with premature deaths. Moreover, present regulatory levels of emissions will necessarily become much stricter.

**Keywords:** steel; environment; mining; production; circular economy; lean and frugal design; ecology transition; climate change; pollution; toxicology

## 1. Introduction

The present article discusses the connection between the environment and steel, both production and use. It argues about the sustainability of steel as a material, and more precisely discusses the triptych: Society, the environment, and materials [1].

To define the topics that ought to be covered in this review, we refer to a simple ecological model of the planet, the *spheres model*, which distinguishes between the geosphere, the biosphere, and the anthroposphere [2]. The environment consists of the concatenation of the biosphere and geosphere.

An activity like steel, firmly anchored in the anthroposphere, interacts with:

- The geosphere because raw materials and energy resources stem from there and much of the waste generated by mining, industry, and end-of-life returns there;
- The biosphere because emissions to air, water, and soil leak into the atmosphere, the hydrosphere, and the top of the geosphere. This should also include Greenhouse gas (GHG) emissions. Steelmaking activities pollute and affect ecosystems and biodiversity;
- The anthroposphere itself, because steel-related elements and compounds influence the health of people (toxicological effects) while steel-related activities create economic and social benefits, directly through the use of steel in the anthroposphere, or indirectly, through its eco-socio-systemic services [3]—two sides of the balance sheet with the pros and cons of steel in the economy and society.

Steel ought to be appraised not simply through its direct production, i.e., through the activities of the steel industry, but through its whole value chain, from raw materials to consumer and investment goods, as well as through its lifecycle, thus tracking the material in goods until and including end of life and its involvement in the circular economy. This is similar to a Life cycle assessment (LCA) approach.

Finally, the theme is time dependent, with a strong historical dimension, both in the long term and short term.

This review is original in the way it brings together the complex set of environmental issues as exhaustively as possible, by going beyond the steel mill itself and into the whole value chain of steel, thus mining, use of steel, pollution, reuse and recycling and health matters, etc. It also addresses the most pressing contemporary environmental topics, like climate change and air pollution, while the discussion stresses the role of society in framing issues and looking for mitigation solutions.

## 2. Steel and Raw Materials

The oldest environmental issue related to steel is the matter of its resources.

Historically, iron ore was ubiquitous in all the parts of the world and, therefore, steel production was located near energy resources rather than ore deposits, thus near water streams for hydraulic power, then near forests for access to large quantities of charcoal, and, since the use of earth coal, near coal mines. The 20th century saw the "discovery" of larger deposits, capable of mines that would be exploited for a long time. In the second half of the century, the scales were turned around with the discovery of high-grade ore deposits (almost pure hematite, more rarely magnetite) and the buildup of a logistical chain based on gigantic ore carriers and new high-capacity export and import harbors. The steel business moved to large sea harbors or to very large waterways inland, like the Great Lakes, the Mississippi, the Rhine, the Danube, or the Yangtze rivers. This was organized with economic targets in mind, based on the rationale of economies of scale, as, indeed, the integrated steel mills grew in size accordingly while the steel market exploded: this was the first wave of globalization, before a second wave of globalization moved goods around the world, beyond raw materials.

Assumed resources of iron ore, worldwide, ought to be able to meet demand for years to come, a cautious expression that should be taken to mean indefinitely—indeed, according to the USGS, world reserves of iron ore were estimated for 2018 at $170 \times 10^9$ tons, containing $84 \times 10^9$ tons of iron and resources at more than $800 \times 10^9$ tons of crude ore containing over $230 \times 10^9$ tons of iron [4]. Taking reserves into account strengthens the conclusion that it not a critical raw material, based on a definition of criticality close to that of scarcity. However, one can distinguish between physical *long-term scarcity* and economic *short-term scarcity* [5]. While there is no risk of long-term scarcity, the discrepancy between the mining and steel sectors' characteristic business times (typically, 5 and 20 years, respectively) may create a short-term scarcity when steel demand is high while the ore offer/supply from exploited mines has not caught up yet, such as in 2007 and 2008 prior to the economic crisis, and, therefore, price excursions may occur [6].

Note also that the circular economy is already changing the deal, as large quantities of recycled material (36% worldwide) feed the steel business as a secondary raw material in parallel with the primary one [7].

Among the other raw materials that the steel sector uses, coal is not critical either, although coking coal was put on the European list of critical raw materials [8], and neither is lime. Some alloying elements may be considered as critical, such as vanadium for example [9].

Producing a material like steel involves engaging much larger quantities of raw materials than the crude steel produced: the "useful" element of economic value, iron, is concentrated down the process route while the unneeded material is sent back to the "environment", usually to landfills, and is labelled as waste.

Figure 1 shows a typical plot of the mass engaged for the production of steel in an integrated steel mill, from the mine to the exit of the steelmaking shop, where crude steel is generated [10]: In this example, the ratio of engaged material vs. useful material is 16.5 to 1. It is a fairly typical set of data,

although the upstream figures (here a burden/ore ratio of 3, slightly on the high side) vary with the iron mine.

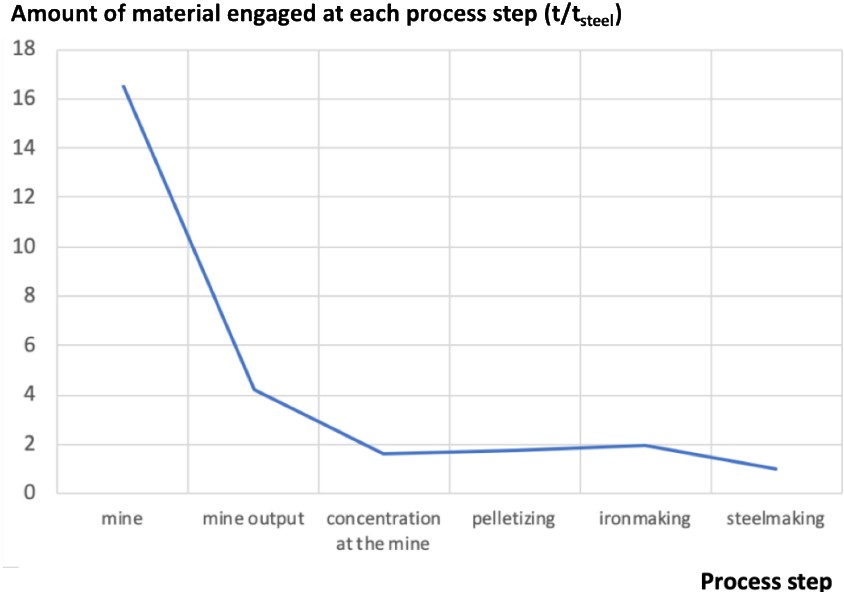

**Figure 1.** Amount of material engaged at each process step (in t). The final output of the process line is 1 t of crude steel (source: author).

Note that waste is generated mainly at three steps, cf. Figure 2. At the mine, the overburden (cf. Figure 3) represents the largest amount of waste (12.3 t); it corresponds to what is left once the ore proper has been separated. It is thus physically identical to the rock present in the mine, minus the ore. Further, in the mining facilities, during the process of beneficiation, a second type of waste is generated after the crude ore is crushed and only the larger-sized ore is retained while the rest, of smaller size, is washed away with water. The output is a slurry called tailings or tailings fines (2.6 t), sometimes contaminated with mineral or chemical additives: It is useless as an iron source with present beneficiation and steelmaking technologies, although it contains iron at the level of a low-grade ore. The third kind of waste is generated in the steel mill itself. It is composed of slag, dust, and millscale, simply called slag here (0.385 t).

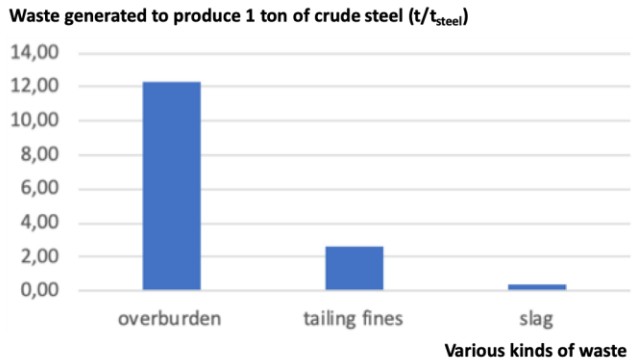

**Figure 2.** Waste generated in the production of 1 t of crude steel (source: author).

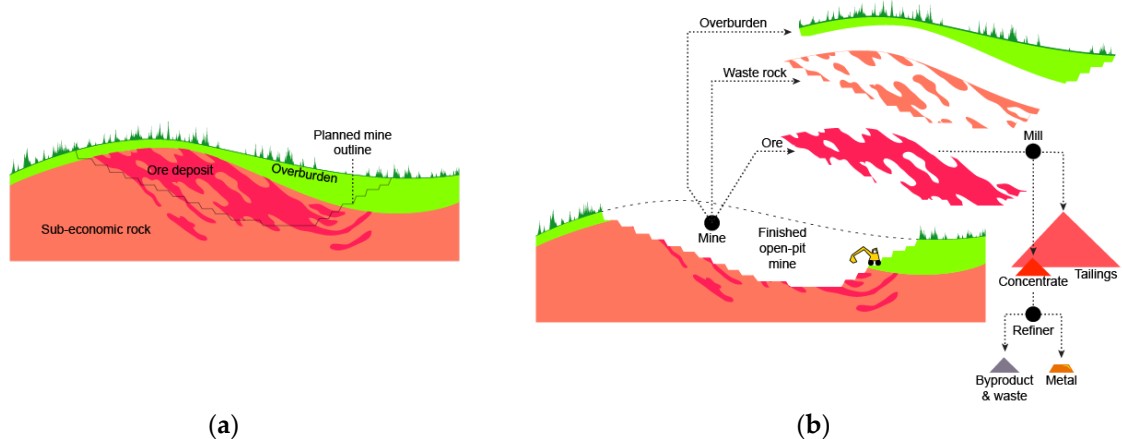

(**a**)                                                    (**b**)

**Figure 3.** Basic mining and metal refining terminology (adapted from groundtruthtrekking.org). Before (**a**) and after (**b**) starting mining operation.

Tailings raise environmental issues because they are often contaminated and are usually stored as slurries in tailings ponds confined behind a dam or an impoundment—in 2000, there were about 3500 active tailings impoundments in the world.

Tailings dam failures are a major risk and have led to numerous mining disasters [11–13]. There were 31 recorded major failures between 2008 and 2018, and 5 already took place in 2019, including the Brumadinho disaster in Brazil shown in Figure 4. Dam disasters have occurred in connection with the mining of all metals beyond iron. In addition to claiming lives, they pollute waterways, land, and ecosystems. The cost of dam failures to the mining business is also high.

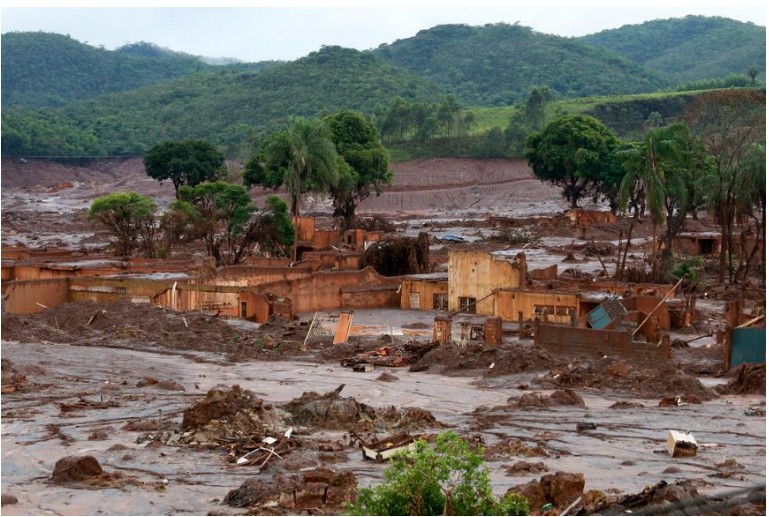

**Figure 4.** Brumadinho dam disaster, which occurred on 25 January 2019 when Dam I, a tailings dam at the Córrego do Feijão iron ore mine, collapsed.

Solutions to avoid dam failures have been identified, including the retrieval of tailings, after draining most of the water from the slurry, for example, by using filter presses. The dried slurry can be reused as building material, for example, in bricks. If the tailings are laid over natural ground, phytoremediation can be used to clean up the new soil, anchor it on the stable substrate, and thus avoid mud slides.

*Dam failures are one of the major environmental liabilities of the mining industry and therefore also of the production of iron, as mining is part of a value chain, which in effect shares negative and positive burdens.*

These various issues ought to be included in an LCA, a method that aims at giving a full picture of the iron lifecycle, from cradle to grave. The dam failure issue, however, is not included in a standard LCA. Furthermore, the full upstream mass balance related to raw materials may or may not be included, depending on the scope of the study (gate or cradle) and on whether the overburden is properly taken on board or not. Therefore, some reexamination of LCA methodology or the development of some other metrics ought to be considered in the future.

## 3. Steel and Energy

The steel industry is considered as an *energy-intensive industry*, especially since *energy conservation* and *climate change* issues have created a drive towards the energy and ecological transitions. This can either be a tautological point, as making steel from ores requires a minimum amount of energy set by thermodynamics, or it can point to inefficiencies in industrial processes, which can and ought to be corrected. Since Roman times, the energy efficiency of carbon-based iron ore reduction was improved by a factor of 100, roughly. Today, the best performing steel mills are within 10% or 15% of a technical optimum [14], although not of the thermodynamic limit.

The steel sector, today, uses coal, natural gas (NG), and electricity mostly as energy sources, depending on the processing route, i.e., either the integrated route, the direct reduction route, or the electric arc furnace route, respectively.

Historically, however, iron was produced entirely from "renewables", either biomass in the form of *charcoal* for the bloomery, the muscles of blacksmiths, or hydraulic power (water wheels) to power the forge. The switch to coal and coke took place in the 18th century and is considered as one of the markers of the *first industrial revolution*. At the beginning of the 20th century, the generation of electricity moved iron production further away from renewables.

The *change from charcoal to coal* took place after a major environmental crisis, when the demand for charcoal contributed to forest depletion in industrialized countries [15]. Let us remember that the "discovery" of coal took place at a time when wood was becoming scarce, an early example of *material criticality* and *anthropogenically induced scarcity*! It also shows that *technology is a social construct*:; coal, which had been around forever, was "discovered" when it was needed by society.

Energy conservation in the steel sector was driven by the fact that energy costs account for roughly 20% of operating costs and therefore needed to be minimized for sound management. Steel therefore was one of the first industries to react to high energy prices, ever since the first energy crisis of 1974, cf. Figure 5. It is not simply a story of energy conservation, however, as the process chain for making steel matured and was perfected in the late 20th century and therefore made it easy to capitalize this cumulated improvement—although abruptly and this caused social pain in steel-intensive regions in the world. Last, this shows that energy conservation is one of the environmental constraints that has been internalized early in the market economy in which the steel sector functions, contrary to the usual paradigm, whereby the environment is considered as an externality.

An estimate of the energy consumption of a best-run integrated steel mill is 18.83 GJ/$t_{HRC}$ (per ton of hot rolled coil). The corresponding mill is shown in Figure 6, complete with a detailed mass and energy balance at each process step. The energy balance refers to the steel mill, gate to gate (ore and coal in, hot rolled coil of steel out). An electric arc furnace (EAF) steel mill consumes 4.29 GJ/$t_{HRC}$ (cf. Figure 7) and a direct-reduction and EAF-based mill 15.6 GJ/$t_{HRC}$ (cf. Figure 8).

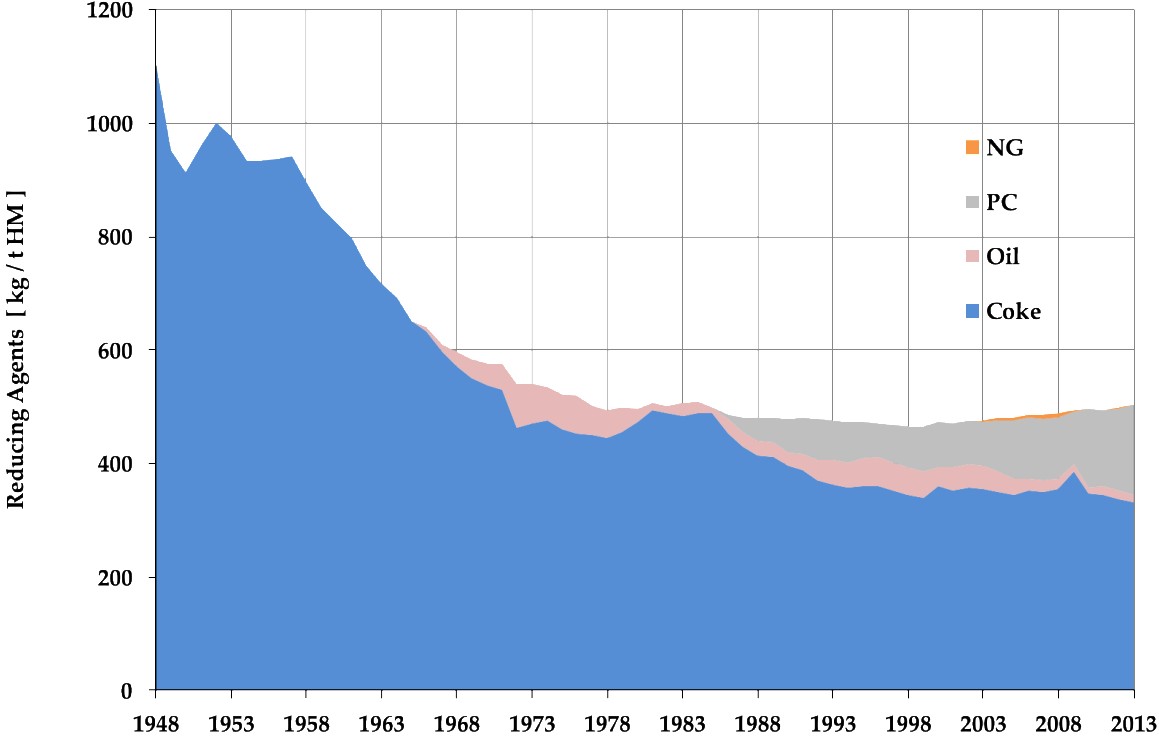

**Figure 5.** Historical evolution of reducing agent's consumption in the blast furnace (BF) in Europe (European Blast Furnace Committee, courtesy of VDEh).

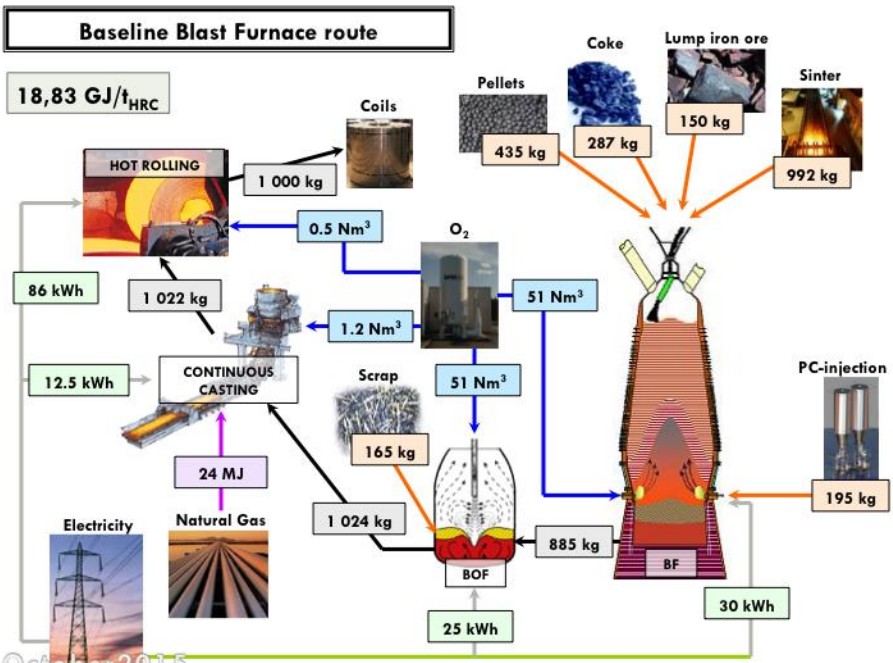

**Figure 6.** Mass and energy balance in an integrated steel mill (baseline blast furnace route, ULCOS simulation). Energy per ton of hot rolled coil (HRC).

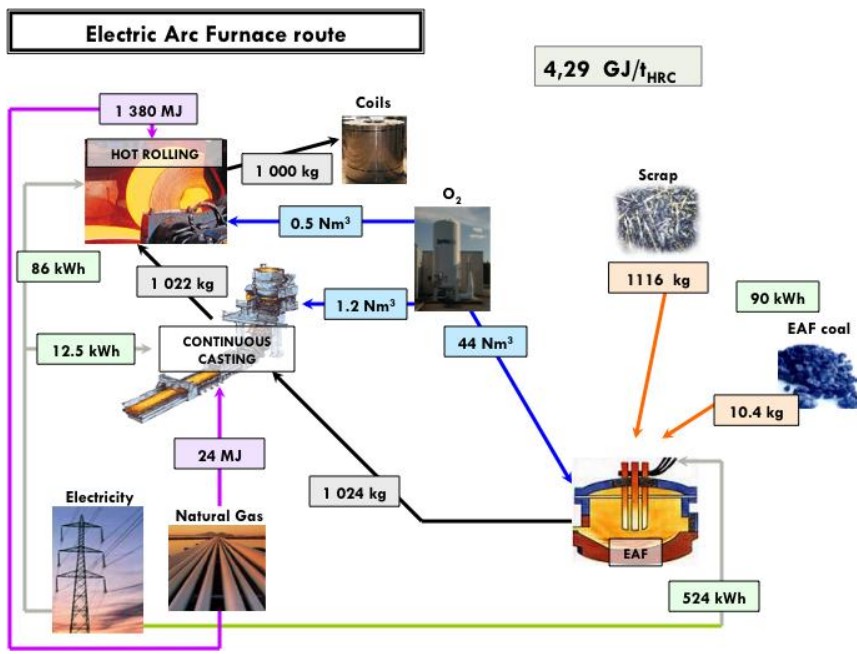

**Figure 7.** Mass and energy balance in an EAF steel mill fed with scrap.

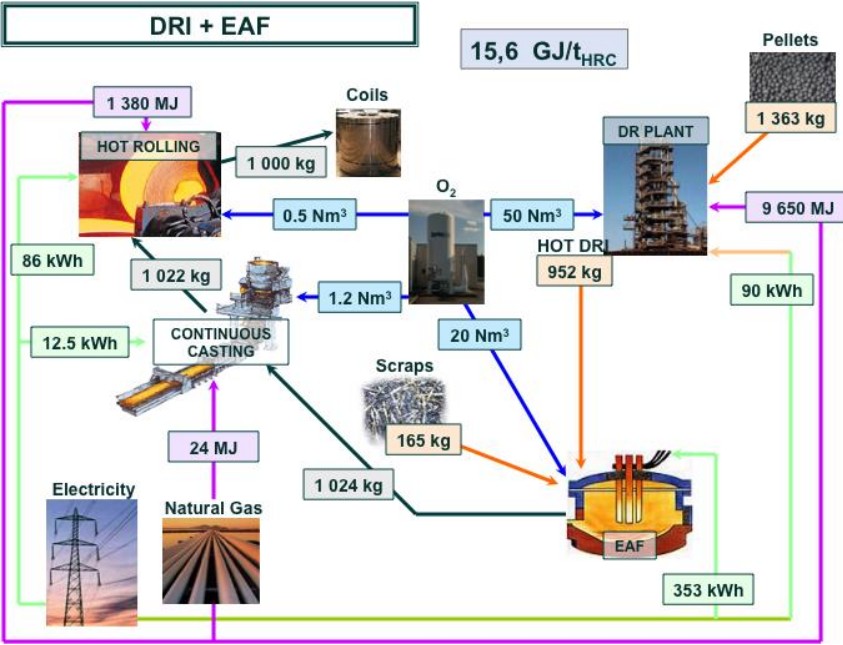

**Figure 8.** Mass and energy balance in an EAF-based steel mill fed with hot Direct Reduced Iron.

R&D into low-carbon steelmaking conducted as part of the ULCOS program (Ultra-LOw $CO_2$ Steelmaking) demonstrated that changing the operating point of the blast furnace and of most of the "ULCOS solutions" made it possible to improve energy use by roughly 20% to 25%, and not simply the 10–15% improvement stated before [14]. This is an extra benefit to be collected from switching to low-carbon process routes.

Hic et nunc, the energy transition, pushed by the cost of energy, by climate change policies, and by the perceived short life of the fossil energy resource (peak oil and peak gas), has pushed steelmaking processes to decarbonize. However, since energy consumption and GHG emissions have decoupled

and will continue do so in the future, part of the story of upcoming progress in the area will be told in Section 5.

This, however, raises a number of interesting issues.

First, what role can electrification play in increasing the use of renewables in steel production?

Electricity is the simplest way to integrate renewables in the energy system. This is done either by injecting them directly at the level of the grid or by trading green certificates. In the short term, electricity is used in the steel sector by *electric arc furnaces* and all the electrical equipment used in a steel mill. In the future, it might be used to electrolyze water and *generate hydrogen*, to be used thereafter for *direct reduction of iron ore,* or to power the *direct electrolysis of iron ore*. Other processes, like the *reheating of steel* for hot rolling or heat treatments, can also be performed in electric heating furnaces (induction, conduction). A lot of technology is available but the high price of electricity, until now, curbed its broad use. There is therefore much leeway left to introduce more electricity in steelmaking and thus to decarbonize the sector at the same pace as electricity decarbonizes.

Note, however, that the energy needs of a steel mill are very large in terms of electrical power: A 5 Mt/y steel mill based on ore electrolysis would require a 1200 MW nuclear power plant or 240 recent wind turbines. This might require new investments in power generation [16]. On the other hand, this would open up opportunities in terms of *demand-side management* of the electricity grid. Indeed, if the steel mill can be turned on and off to accommodate electricity demand, this would alleviate or even suppress the complex matter of dealing with the intermittency of renewable electricity [17].

Second, are there other ways than green electricity to use renewables in steel production?

The short answer is no, as renewables are not meant to supply high-temperature energy of the kind that the steel industry needs. Heat reduction is out of reach, cf. Section 5.2. This should not keep inventors from trying to find new solutions, however.

Third, would not this decouple the search for less GHG emissions from the doxa of minimum energy consumption?

The implicit assumption today is that energy should be optimized first and then GHG emissions in a second step. If the price of carbon increased enough, these priorities would switch. The change could also be a matter of policy. This would release many constraints in the search for low-carbon solutions. Note, incidentally, that introducing renewables in any industrial system implicitly negates the energy optimization rule: Indeed, renewable energy is a rather inefficient way of generating electricity and, likewise, producing biomass by photosynthesis is also very inefficient, but this does not eliminate it from the search for solutions.

Finally, the point may not be so much to decrease the energy intensity (J/kg) of making steel, which may have reached a physical limit in the best-operated steel mills, but to decrease the lifecycle sector's or the value-chain's overall energy consumption. This will be achieved by *lean* and *frugal practices*, including reuse and recycling and *product-service systems (PSSs)*. Indeed, using less steel for the same services is a solution to cut energy consumption and GHG emissions at the same time. Sharing a car or a car ride, or an apartment in a short-term rental scheme are part of these solutions—this, in effect, increases the "productivity" of cars of or homes, minimizes the amount of steel engaged per unit of service, and should therefore decrease their environmental footprint, provided the extra management cost/footprint of these services remains small enough.

The future of energy is therefore deeply related to low-carbon practices at the level of large systems like the steel sector, on the one hand, and of individuals and their lifestyles, on the other hand.

## 4. Air Emissions and Pollution Related to Steel

*Anthropogenic activities* generate emissions to the atmosphere (air), the hydrosphere (water), or the geosphere (ground): The *emissions* evolve, interact, concatenate, sometimes move from one environmental compartment to another (e.g., ground to water), and eventually end up constituting what is called *overall pollution*. Thus, one should speak of the emissions of road traffic but of air pollution in a city and a region. A steel mill emits to the three environmental compartments and

these emissions add up to the other local emissions. Another level of complexity is related to the geographical scale of the territory affected by that pollution: Emissions start as a local phenomenon, but pollution may propagate to other regions, countries, and sometimes continents. Thus, for example, *dioxin emissions* are local while *greenhouse gas emissions* are truly global, at the level of the whole planet. The trend, since the turn of the century, has been for more and more emissions to become more and more global.

Emissions are analyzed at a steel mill's or a single reactor's scale, either as statistical data on emissions or as process engineering analyses of how the emissions are generated. The global phenomenon of atmospheric pollution is only analyzed by meteorological tools. Emissions of the value chain, especially of its upstream part, should also be part of the discussion.

### 4.1. Air Emissions and Pollution Related to Steel

Data on the steel sector's air emissions are available from statistical data [18] and from publications about *the best available technologies*. They are expressed as emissions factors, either per ton of steel $(g/t_{steel})$ or per volume of smokestack emissions $(mg/Nm^3)$—they can be converted into each other, using the data published in [19].

Examples are given in Table 1 [19] and Table 2 [20], where different emissions parameters are presented: Heavy metals or HMs (Pb, Cd, Hg, Cr, Cu, NI, Se, Zn plus As, a non-metal), $SO_x$, $NO_x$, organic compounds (NMVOC, PCB, PCDD/F, PAHs, HCB), and particulate matter (PM, $PM_{10}$, $PM_{2.5}$, TSP); the first table shows global industry-wide figures while the second gives details of the emission limits of individual plants in the steel mill.

**Table 1.** Steelmill specific air emissions of heavy metals and organic compounds, sector averages and spreads [19].

| Pollutant | Value | Unit | 95% Confidence Interval | |
|:---:|:---:|:---:|:---:|:---:|
| | | | **Lower** | **Upper** |
| NMVOC | 150 | $g/Mg_{steel}$ | 55 | 440 |
| TSP | 300 | $g/Mg_{steel}$ | 90 | 1300 |
| $PM_{10}$ | 180 | $g/Mg_{steel}$ | 60 | 700 |
| $PM_{2.5}$ | 140 | % of $PM_{2.5}$ | 40 | 500 |
| BC | 0.36 | $g/Mg_{steel}$ | 0.18 | 0.72 |
| Pb | 4.6 | $g/Mg_{steel}$ | 0.5 | 46 |
| Cd | 0.02 | $g/Mg_{steel}$ | 0.003 | 0.1 |
| Hg | 0.1 | $g/Mg_{steel}$ | 0.02 | 0.5 |
| As | 0.4 | $g/Mg_{steel}$ | 0.08 | 2.0 |
| Cr | 4.5 | $g/Mg_{steel}$ | 0.5 | 45.0 |
| Cu | 0.07 | $g/Mg_{steel}$ | 0.01 | 0.3 |
| Ni | 0.14 | $g/Mg_{steel}$ | 0.1 | 1.1 |
| Se | 0.02 | $g/Mg_{steel}$ | 0.002 | 0.2 |
| Zn | 4 | $g/Mg_{steel}$ | 0.4 | 43 |
| PCB | 2.5 | $mg/Mg_{steel}$ | 0.01 | 5.0 |
| PCDD/F | 3 | $\mu g$ I-TEQ/$Mg_{steel}$ | 0.04 | 6.0 |
| Total 4 PAHs | 0.48 | $g/Mg_{steel}$ | 0.009 | 0.97 |
| HCB | 0.03 | $mg/Mg_{steel}$ | 0.003 | 0.3 |

**Table 2.** Target values of various specific air emissions factors for steel production processes [20].

| Pollutant | Process | Limit Values (mg/Nm$^3$) | | | | |
|---|---|---|---|---|---|---|
| | | HM Protocol 1998 | Gothenburg Protocol 1999/2005 | Gothenburg Protocol 2012 | HM Protocol 2012 | POP Protocol 2012 |
| SO$_x$ | combustion of coke oven gas | - | new: 400 existing: 800 | 400 | - | - |
| | combustion of blast furnace gas | | new: 200 existing: 800 | 200 | - | - |
| NO$_x$ | combustion of other gaseous fuel | - | new: 200 existing: 350 | new: 200 existing: 300 | - | - |
| | sinter plant | - | 400 | 400 | - | - |
| Particulate matter | sinter plant | 50 | - | 50 | | - |
| | pelletizing plant | 25 40 g/t$_{pellets}$ | - | crushing, grinding & drying: 20 all other process steps: 15 | | - |
| | blast furnace | 50 | - | hot stoves: 10 | | - |
| | basic oxygen steelmaking | - | - | 30 | | - |
| | electric steelmaking | 20 | - | new: 5 - existing: 15 | | - |
| | hot & cold rolling | - | - | 20 bag filters not applicable: 50 | | - |
| PCDD/F | sinter plant | - | - | - | | 0.5 ng/m$^3$ |
| | electric arc furnace plant | - | - | - | | 0.5 ng/m$^3$ |

The list of pollutants mentioned in these tables is far from complete. Pollutants missing are, for example, silica (work on refractories), asbestos (not used any more, in most countries), and acid fumes (on pickling lines), etc. Moreover, categories inherited from toxicology (cf. Section 7) are not used in these inventories, e.g., carcinogenic or endocrine disruptors, although they would be relevant. Note also that some extremely important emissions could not be easily measured until rather recently (dioxins, 20 years ago, PM$_{2.5}$ and PM$_1$, 5 years ago and PM$_{0.1}$, presently).

The data on emissions are relative to industrial equipment, therefore to the combination of process reactors and exhaust gas-cleaning devices. Direct emissions from the processes alone are one or two orders of magnitude larger than what is presently dispersed to the atmosphere.

The ironmaking plant, comprising coke ovens, a sinter plant, and pelletizing plus a blast furnace, has the reputation of being the largest emitter in terms of particulate matters and combustion, although this "order of merit" can change depending on the actual technology used and on its age. Note also that emissions from ore piles, which are stocked inside the steel mill upstream of the ironmaking plant, are not usually accounted for in these emissions, even though this may raise serious issues (cf. Section 7).

*4.2. Dust Generation*

A more phenomenological approach can be adopted to identify the mechanisms that generate the dust [21], cf. Figure 9:

- *Saltation*, the basic mechanism for transporting powder from piles by the action of wind or of gas convection. It is the basic mechanism responsible for the airborne dust originating from sand deserts and transported thousands of kilometers away and for the displacement of sand dunes as well [22]. It is also effective in eroding piles of raw material, like ore, for example, and projecting dust in the atmosphere.
- *Erosion* can also generate dust, a common phenomenon in the biosphere/geosphere.
- *Volatilization (evaporation)* of volatile species (arrow 1 in the figure), such as some metals (e.g., zinc, lead, silicon—even though it is not a metal). This happens in the EAF and is the major mechanism in ferro-silicon furnaces (see further).
- *Projection by gas, electric arc, or powder injection* (arrows 2 and 2′), as in an EAF where coal or lime is transported with gas into the liquid steel bath through immerged lances while oxygen is injected in the same manner.

- *Bubble burst* (arrow 3 and 4) is active when large amounts of gas are generated in a liquid bath reactor, either by injection or as a result of a chemical reaction, like the decarburizing of the bath by "burning" dissolved carbon by injected oxygen.
- Probably more, like the *reemission of incoming* dust (arrow 5) or the *precipitation of graphite* when tapping a hot metal (pig iron) ladle or just letting it just sit while waiting for the converter.
- Note a variant of the bubble burst, called *droplet burst* [23], a phenomenon due to the oxidation of the droplet, its encapsulation as liquid metal inside a rigid shell, and then the explosion of that shell due to the buildup of inside pressure by evaporation, followed by the explosion of the droplet itself. Evidence of this mechanism was reported by Nedar Lotta in a steel BOF, but not by Huber et al. in the EAF. It is also reported in a different context, that of the explosion of fuel droplets during combustion.

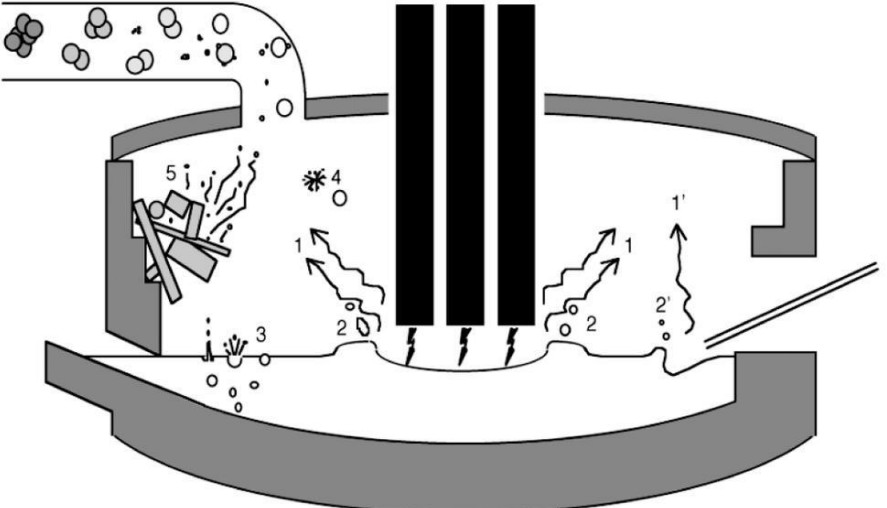

**Figure 9.** Mechanisms of dust generation in an electric arc furnace [21].

These various mechanisms lead to the formation of dust of a different nature in an electric arc furnace, as shown in Figure 10.

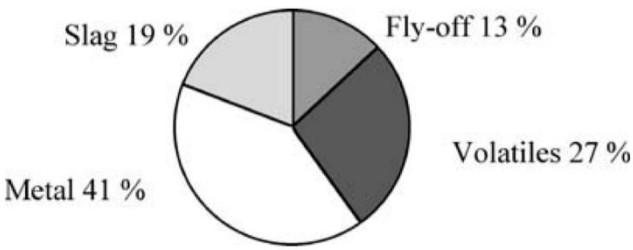

**Figure 10.** Nature of particulates generated inside an EAF [21].

*Bubble burst* was identified as the main mechanism responsible for the generation of dust in a steel electric arc furnace [24–36], cf. Figure 11. Experimental evidence in a steel bath is also shown at the top of that figure.

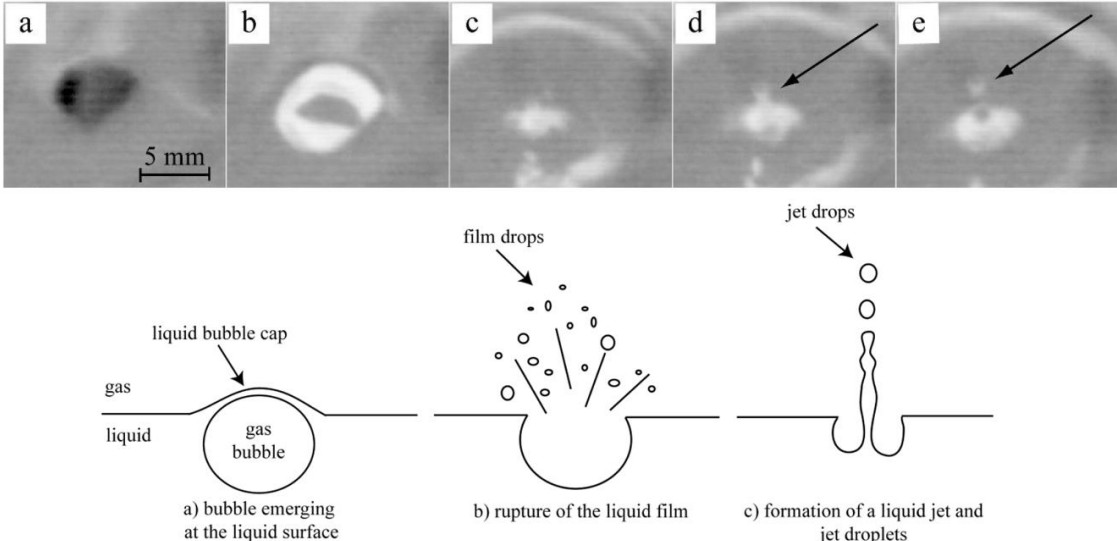

**Figure 11.** Top: frames taken from a high-speed video (emergence of a bubble (**a**,**b**), disruption of the bubble cap (**c**), formation of vertical liquid jet (**d**), emission of a drop from the jet (**e**); Bottom: bubble burst mechanism for dust generation [36].

Gas bubbles, generated in the melt by the nucleation of gaseous CO, rise to the free surface, where they explode and liberate small droplets of liquid iron, either from the film of the bubble or from the column of liquid metal that rises in the center of the bubble when the film collapses, the latter contributing more than the former; the droplets are then sucked into the gas exhaust system, where they solidify and oxidize in the draft alongside zinc vapor, which is the other major contributor to dust generation and originates from the evaporation mechanism.

The proportion of iron involved in an EAF due to this mechanism is 10.5 $kg_{dust}/t_{steel}$.

Dust is collected in a dry or a wet system, on the EAF itself (primary collection) or under the canopy of the steel shop (secondary collection), and generates the powdered dust or the sludge by-product. Some dust bypasses the collection systems and gets dispersed in the atmosphere to eventually drop to the ground (<5%).

Bubble burst is active in all liquid metal reactors, like converters or ladles. In combination with gas or powder injection, it is also essential in a liquid bath smelter.

*Evaporation* is the main mechanism in reactors that are "quiet", i.e., not stirred by powerful injections, decarburizing, or other mechanisms of in-bath gas generation, and which handle materials with a high vapor pressure. Metal is often the main substance that directly evaporates, like zinc in a steel EAF, or at the top of an imperial smelting furnace (ISF). Dust is only generated if at some point the vapor comes in contact with oxygen from air ingress; otherwise, the metal will condense and be recovered as in the ISF.

In the case of the production of *high-silicon alloys* including ferro-silicons, the volatile substance that escapes the furnace is SiO and not silicon itself [36]. Figure 12 shows the major mechanisms of dust generation in this context.

Evaporation is used to handle *dust from steel EAFs*, which are hazardous by-products and as such need to be treated separately. The Waelz kiln's process is based on the evaporation of zinc. Similarly, steel producers sometimes recycle the dust in the EAF itself, and thus enrich it in zinc by evaporation, up to 20% or 30%, in order to reduce the fee that they have to pay to the dust processors, which does not necessarily make economic sense from a total *cost of ownership* standpoint [37].

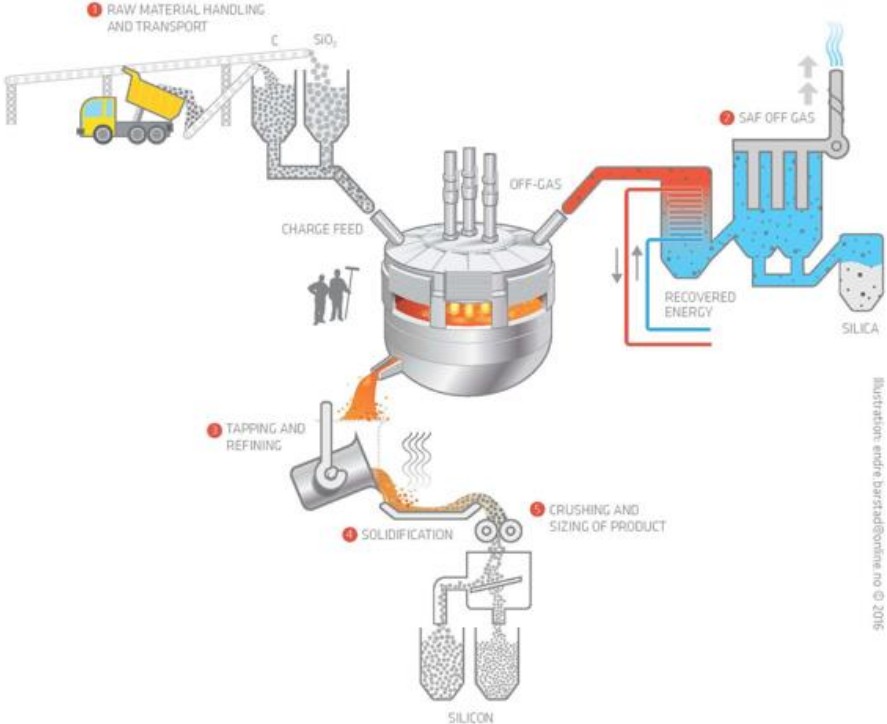

**Figure 12.** High silicon alloy production process and its primary emission sources [36].

## 4.3. Organic Compounds Genesis and Emissions

The *generation of organic emissions* is another area where process mechanisms have been described.

Studies of the emission of organic compounds by industrial processes are basically of two types: On the one hand, measurement campaigns are carried out to assess emissions both in terms of compounds formed and of their concentration [27]—this is a necessary step to analyze the compliance with regulatory limits on emissions; on the other hand, more basic work was carried out to understand the physico-chemistry mechanisms that drive the generation of some particular molecules from some particular sources [38–43].

For example, the transformation of materials added as contaminants to commercial scrap to the EAF charge, for example, bits of car tires, was analyzed in a laboratory simulator mimicking various conditions within the reactor. The degradation of the organic contaminant is due both to pyrolysis and to oxidation reactions; their relative importance depends on the temperature and on the concentration of oxygen.

Under pyrolytic conditions, hydrocarbons are thermodynamically unstable above 500 °C. Generally, their stability decreases when the molecular weight increases. Methane is the most stable compound below 1000 °C. A double bond is more stable than a single one. Because of resonance energy, aromatic compounds are very stable at high temperatures. Compared to other hydrocarbons excepting aromatic compounds, alkynes are more stable between 1000 and 1200 °C.

Therefore, under oxidizing conditions, both pyrolysis and oxidation take place, leading to the emission of numerous molecules more or less oxidized. Final products like carbon dioxide and water, inert gases but also unburnt compounds, such as $H_2$, CO, tars, VOCs, PAHs, etc.

More complex and harmful molecules can be generated in or outside of process reactors (for example, in their fume collection system), like *tetrachlorodibenzo-furans (PCDF)* and *tetrachlorodibenzo-dioxins (PCDD)*, both usually simply referred to as *dioxins* [38,39,43,44]. These compounds have a bad reputation in terms of health consequences (cf. Section 7) and therefore have been chased relentlessly by regulators. Industry has adapted accordingly.

Dioxins tend to form in an oxidizing atmosphere, thus rarely in the laboratory of the process reactor but most commonly in fume collection systems. Thus, in a steel blast furnace, which operates under reducing conditions, dioxins are never encountered. On the other hand, sinter plants, EAFs, and municipal waste incinerators (MWIs) have been well-known sources of dioxins.

Dioxins are generated by the degradation of organic compounds present in the charge of the reactor, where they are introduced either deliberately (fossil fuel, biomass) or as contaminants, but also by synthesis from inorganic substances, the so-called *de-novo mechanism*. Dioxins are gases at high process temperatures and then they condense as liquids in the dust collecting system (cf. Figure 13). Thus, for example, EAF dust is heavily contaminated by dioxins, which, in turn, might create problems at dust processors' plants.

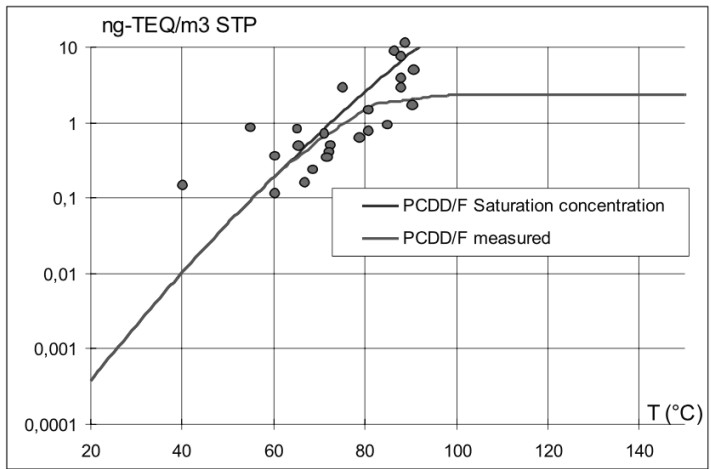

**Figure 13.** Comparison between the TEQ (toxic equivalent) saturation curve and experimental data from Arcelor Luxembourg.

Countermeasures were proposed to make sure that dioxins fully precipitate in the dust and are not carried away in the flue gases, heading for the smokestack and the atmosphere. The preferred abatement mechanism, however, is the post-combustion of the flue gas under conditions where dioxins are properly eliminated—i.e., combusting for at least 2 s and then quenching the gas very quickly, for example, between 400 and 250 °C—sometimes with the boost offered by a catalyzer.

### 4.4. Emissions and Pollution

The contribution of the emissions from a steel mill to pollution is case dependent. For example, the steel mill may be isolated, embedded in a city or part of an industrial region, where several other emitters coexist.

A case study [45] is shown in Figures 14 and 15. It relates to the Metropolitan Region of Vitória, Brazil, on the Atlantic coast, where dust pollution in terms of Total Suspended Particles (TSP, macroscopic dust) and $PM_{10}$ (Particulate Matter less than 10 μm in size, breathable dust) were measured for 5 years (1995–1999) at four or two locations, respectively, for periods of 24 h every week. In total, 15 sources of dust were identified in the region and aggregated into 3 families: Industrial (integrated and EAF steel mills, mining terminal for overseas shipping of iron ore, pelletizing plant, coal, etc.), human activities (forest fires, soils, civil construction, quarries), and natural (marine aerosol). Dust in the Vitória region is overwhelmingly from anthropogenic origin, i.e., industrial and human activities' emissions. Human emissions are always more important than industrial ones, a clear-cut conclusion for large-particle dust and a weaker one for breathable dust.

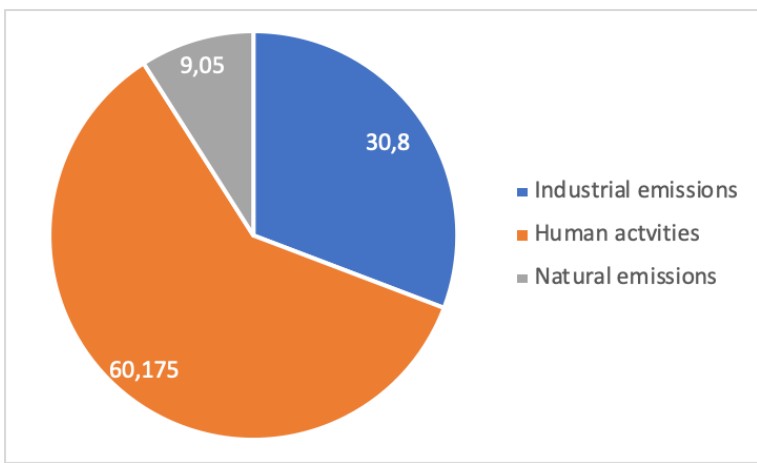

**Figure 14.** Local pollution in the Vitória metropolitan area: distribution of TSP according to sources of emissions (adapted from [45]).

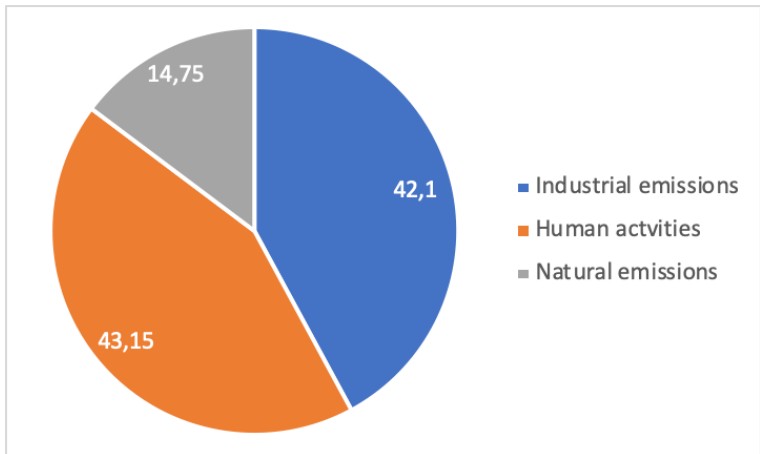

**Figure 15.** Local pollution in the Vitória metropolitan area: distribution of PM10 according to sources of emissions (adapted from [45]).

This shows clearly that pollution is multifactorial, and that industry is not the major emitter in this case, although it is an important one.

Dust pollution is a rather simple pollution phenomenon. Indeed, there are more complex situations where several pollutants add up to create new pollution, like in the case of *photochemical smog*. However, discussing these topics, which are not often analyzed in the literature in connection with steel mill emissions, goes beyond the scope of this article, although it points to interesting areas for further investigations.

Emissions to water and soil ought to be discussed in parallel with the previous discussion focused on air. We leave this for further reviewing work.

### 4.5. Lifecycle Emissions

Emissions from a steel mill, gate to gate, only encompass part of the emissions of the lifecycle of steel. Upstream emissions from mining were mentioned *en passant* in Sections 2 and 4.2 (saltation), relative to iron ore piles, solid waste and dust.

A Life Cycle Assessment (LCA) looks at emissions in a particular way related to *impact categories*. Table 3 from [46] examines seven mid-point impact categories for steel products (beams, coils of strip, etc.), related to: (1) Energy use (*primary energy demand*, PED), (2) GHG emissions (*global warming*

*potential*, GWP), emission of acidic compounds that affect water quality either (3) by turning it acid (*acidification potential*, AP) or by providing nutriment for plant life (4) (*eutrophication potential*, EP), or generating emissions to air that cause *photochemical ozone creation* (POCP).

**Table 3.** Lifecycle significant flows, phases and processes in the worldsteel steel Life Cycle Inventory (excluding the end-of-life phase).

| Impact Category | Main Input/Output | Main Phase | Main Processes |
|---|---|---|---|
| Primary energy demand | Hard coal (75–95%) Natural gas (0–15%) | Upstream (~100%) | Upstream energy: electricity and fuels |
| Global warming potential (100 years) | Carbon dioxide (90–95%) Methane (~6%) | Gate-to-gate (>60%) Upstream (20–30%) | |
| Acidification potential | Sulphur dioxide (50–60%) Nitrogen oxides (30–40%) Hydrogen sulfide (<10%) | Gate-to-gate (40–60%) Upstream (40–60%) | Gate-to-gate: steel production processes up to slab production |
| Eutrophication potential | Nitrogen oxides (>90%) Nitrous oxide (~2%) Ammonia (~2%) COD (~2%) | Gate-to-gate (>60%) Upstream (50–80%) | |
| Photochemical ozone creation potential | Carbon monoxide (60–70%) Sulphur dioxide (10–20%) NMVOCs (<10%) Nitrogen oxides (<10%) | Gate-to-gate (>80%) Upstream (~20%) | |

The second column of Table 3 describes the inputs and the outputs (emissions) in/out of the lifecycle: energy comes from coal or natural gas, GWP is caused by $CO_2$ or $CH_4$ emissions, etc. The third column explains how much of the particular impact category is related to the steel mill itself (gate-to-gate) or to the upstream part of the lifecycle, thus to mining and raw materials' transportation. The upstream part of the lifecycle is significant, as it accounts for 20% to 100% of the midpoint indicator.

Of course, downstream emissions are related in a complex manner to the material. Moreover, beyond the negative impacts of materials, such as air emissions, they also have a positive impact, which has been described as the eco-socio-systemic value of that material [3]. This is, however, going beyond the scope of the present review paper.

There is much room left for more systematic work on emissions. Unifying the vocabulary used by the various disciplines that tackle these matters would be useful and this would make it easier to bring all the data into a single set. More work is needed on the phenomenology of emissions, at experimental and modeling levels. Connection with meteorological pollution and with health issues (see further, Section 7) would further flesh out the overall discussion. Extending the analysis to water and soil would also be necessary. Eventually, when this is all done, a more rigorous review than the present one would be in order.

## 5. Greenhouse Gas Emissions Related to Steel

### 5.1. Anthropogenic vs. Geologic Iron

The stock of iron available for anthropogenic use now and in the future is present under two forms:

- Metallic iron, constituting the stock of steel in the anthroposphere, which comprises steel-in-use engaged in various artifacts and discarded steel, known as scrap or as the anthropogenic mine.
- Iron compounds, mainly oxides, which constitute iron ores and are found in ore deposits (≈resources) and in geological mines (≈reserves).

The *circular economy*, a popular word today but a historical practice in the world of materials, should ensure that the anthropogenic mine takes over 80% of the iron supply by the end of this century [47]. This may read as somewhat paradoxical, as the recycling rate of steel is already above 80–90%. However, the residence time of steel in the economy is of the order of 40 years and the demand for steel is still increasing worldwide so that this complex dynamic will not deliver an intuitively circular economy until then.

Note also that an integrated steel mill, complete with coke oven batteries, generates coke oven gas (COG), rich in hydrogen (roughly 60%), which could serve as a source of hydrogen—and thus combine with low-carbon steelmaking. Such a scheme was indeed put into practice by Air Liquide at the Carcoke company in Zeebrugge, Belgium, in the 1980s. The gas was distributed to various customers through a regional system of pipes, which is still in operation today. The shutdown of the coke ovens brought this production of hydrogen to an end.

*5.2. Reduction of Iron Ore*

Until the advent of a steel production fully based on the circular economy, chemical reduction of iron ore will remain necessary—as physical reduction is out of reach, cf. Figure 16. Indeed, vacuum decomposition is impossible, even in the deep vacuum of outer space ($10^{-11}$ Pa). The best laboratory vacuum (ultra-ultra-vacuum) is only $10^{-10}$ Pa. Thermal decomposition of iron oxides requires very high temperatures (3404 °C), which could be achieved, for example, in solar furnaces. However, as soon as the atoms (ions) would be separated, they would tend to recombine again, unless they are physically kept part, for example, in a kind of mass spectrometer. Needless to say, this does not correspond to a technology commensurate with the mass production of iron! The only physical option available for the decomposition of iron oxide is electrolysis, which will be discussed further in the report.

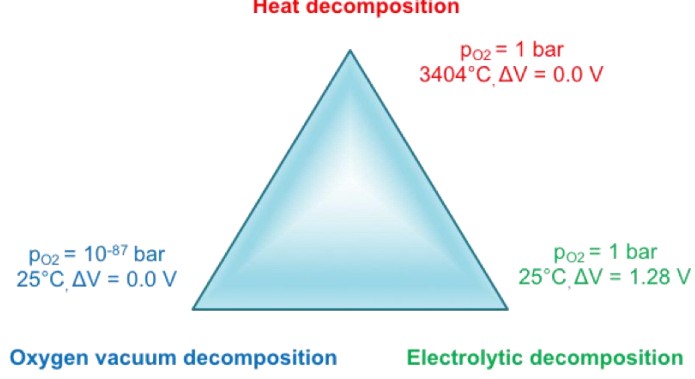

**Figure 16.** How to recover iron from oxides by pure physical means?

This is presently accomplished by smelting reduction based on the use of carbon, mainly from coal and, marginally, from natural gas, and other options include hydrogen and electricity, plus any combination of the three, as schematized in Figure 17, which also shows how they may contribute to low-carbon production.

Basically, there are three ways of reducing iron ore by chemical means:

- Historically, carbon has been used first, as charcoal initially and then as earth coal converted into coke (metallurgical coke). The blast furnace-based, integrated steel mill is the current state-of-the-art technological avatar of this solution.
- Hydrogen is the other option, although it requires the production of hydrogen first, whereas carbon stems directly from mining. Hydrogen reduction has been investigated at the laboratory extensively and some limited industrial operation took places at various scales years ago. See Section 5.9 for the revival of the concept in many on-going research projects.

- Electrons ($e^-$) can also behave as reducing agents, in the technological framework of electrolysis. The method is used extensively in non-ferrous metallurgy, most especially for the production of aluminum. However, several pieces of laboratory work have shown that the concept does work for iron as well (see further details in Sections 5.4 and 5.5).
- In addition to these "mainstream" solutions, there are more, like using metals as reducing agents or powerful reductants like hydrazine ($N_2H_4$). Due to the price of metals, especially non-ferrous metals that would have to be used in the case of steel, such as aluminum, this is not a practical solution, although it is used at the margin, e.g., aluminum is sometimes used in the steel shop to raise steel temperature in the ladle.

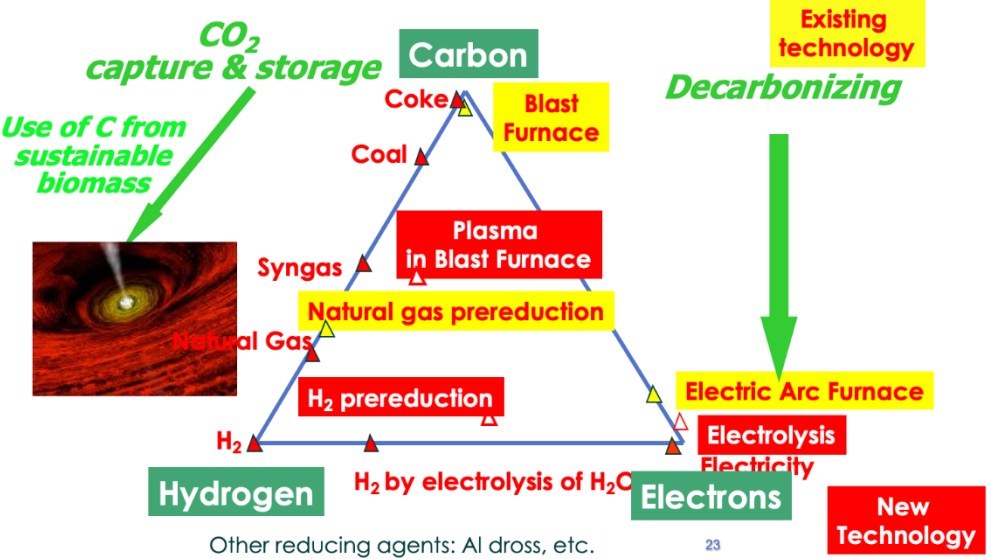

**Figure 17.** Practical ways to reduce iron ores: process route for iron and steelmaking.

Finally, for the sake of completion, hybrid methods have been imagined or tested in a long series of reactor concepts and sometimes pilot plants. An example is the combination of a nuclear power plant, preferably of the fourth generation, with an iron ore reduction plant: the former would provide the latter with part of the energy needed to bring the charge up to the reaction temperature. This work was never published but led to the concept of *high-temperature heat pumps* to boost the temperature above the level that waste heat from the nuclear reactor could deliver.

A more detailed analysis of many more possible ways to make low-carbon iron is available in [48]. It includes in particular a discussion of *thermochemical cycles*.

*5.3. Benchmarking of Various Iron Reduction Routes*

An exhaustive assessment of various existing or new process routes made possible through new technology developments (45 routes, altogether) was carried out in the framework of the ULCOS program [49]. Energy needs, $CO_2$ emissions, and total production costs were evaluated [14,50].

The routes differ slightly in terms of energy needs, but the differences are fairly small: all routes are either already fairly energy efficient (existing routes) or are extrapolated on the basis of high energy efficiency (new routes)—cf. Figure 18. Important details are in the original papers.

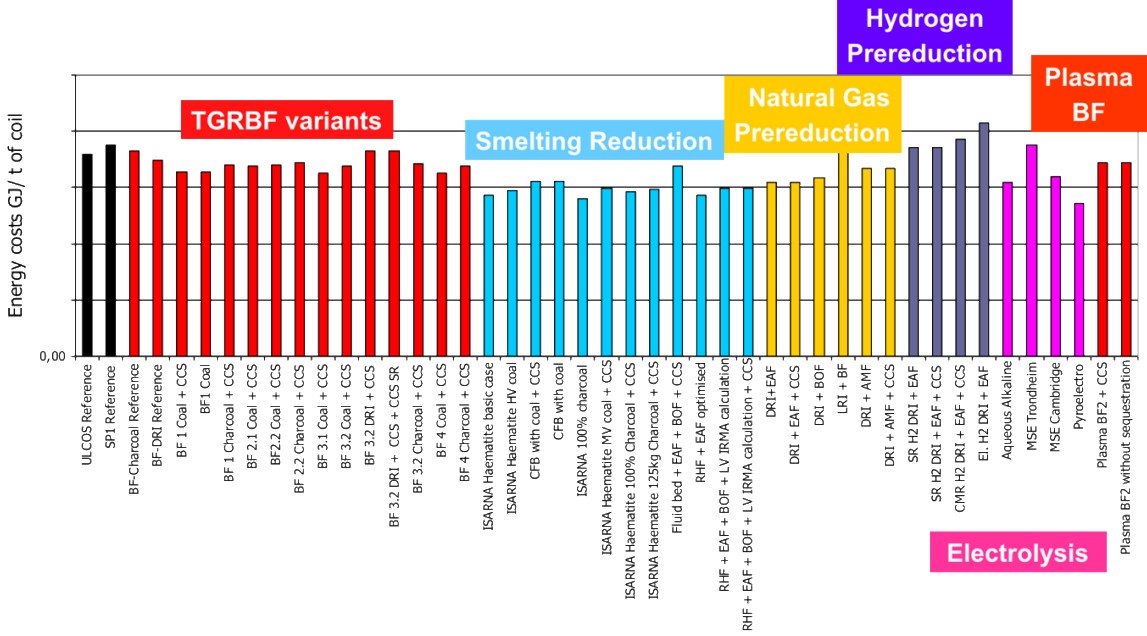

**Figure 18.** Specific energy needs (GJ/t$_{HRC}$).

CO$_2$ emissions, shown in Figure 19, are much more spread, because some of the routes adopt low-carbon technologies, such as CCS (carbon capture and storage) or some kind of green electricity. One route even exhibits *negative emissions*, because it uses biomass carbon (charcoal) from *sustainable plantations* and applies CCS, thus sequestering the CO$_2$ that was pumped from the atmosphere (in the plantation). The graph distinguishes between scope I and scope II emissions, i.e., between direct process emissions from steelmaking and indirect emissions (in green) related in particular to the production of electricity.

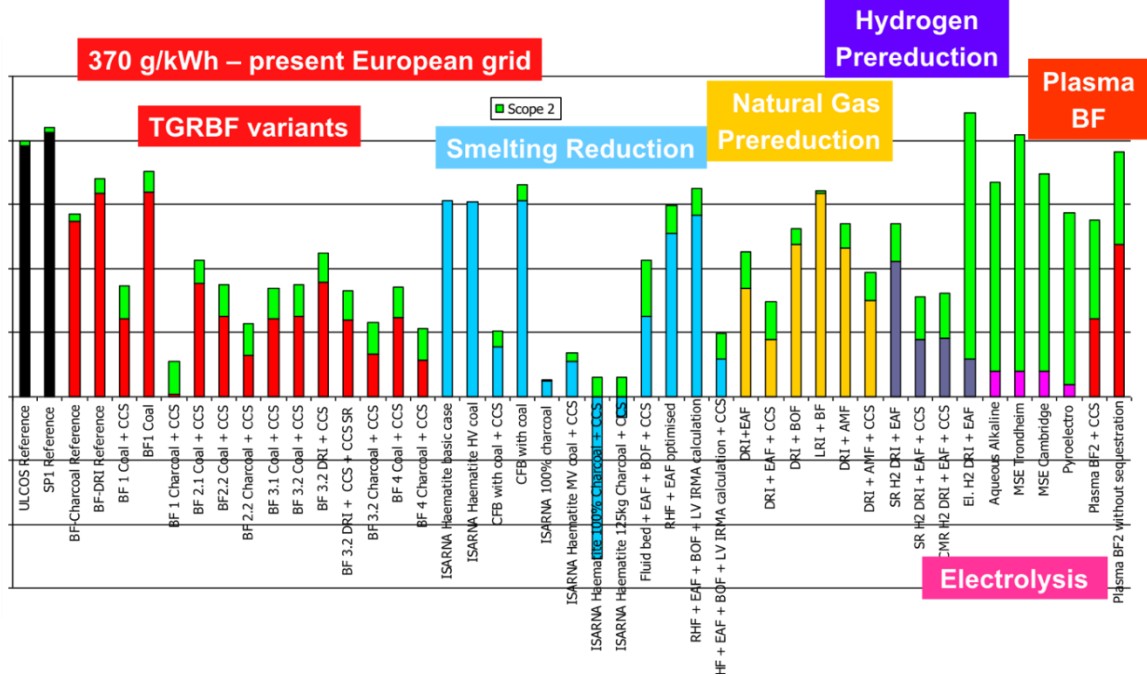

**Figure 19.** CO$_2$ specific emissions (t$_{CO2}$/t$_{HRC}$) for a 370 g/kWh grid.

Paying special attention to the hydrogen routes, they exhibit a slightly higher level of energy need than conventional routes and some other low-carbon solutions but do show improved emissions, except for the electrolysis Direct Reduction (DR) route when the carbon intensity of the electrical grid is 370 g/kWh, which was the European level at the time of the study (cf. Figure 19). The low-carbon grid in Figure 20 shows very low emissions.

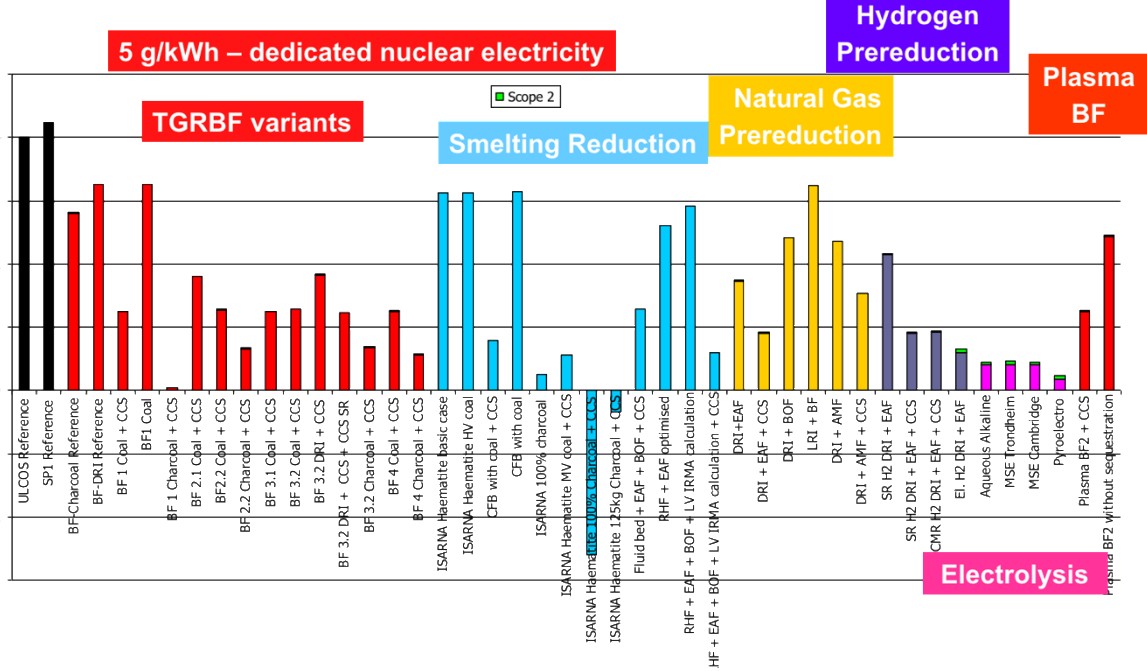

**Figure 20.** $CO_2$ specific emissions ($t_{CO2}/t_{HRC}$) for a 5 g/kWh grid.

The costs, not shown in this summary, are all significantly higher than the production costs of conventional BF ironmaking, which indicates that low-carbon technologies cannot "pay for themselves" and need some kind of extra funding, for example, through a carbon tax.

## 5.4. ULCOS Solutions and Their Present Evolutions

The *ULCOS solutions*, i.e., the short list of process routes that would be pursued at higher TRL after the end of the project, were defined in the second part of the ULCOS program in 2010. They are shown in Figure 21. The first three of them rely on CCS and the fourth one on the direct use of "green" electricity.

*ULCOS-BF* was to be built based on the existing Florange Blast Furnace of ArcelorMittal. It applied to the European NER 300 Program [51] and was to be awarded the required financing, when the blast furnace was shut down because of the economic crisis. The technology has since been frozen, but lower TRL research continued under the names of the VALORCO and LIS programs with French ADEME financing [52]. It is presently awaiting a blast furnace on which to resume the work.

Another ULCOS subproject ran in the first stage of the program (the injection of plasma at the tuyere of the blast furnace in order to reinject CO + $CO_2$ and reduce $CO_2$ with the electrical energy) was re-ignited and is now running as the IGAR project in Dunkerque plant, as schematized in Figure 22; it has received financing from French SGPI as part of the PIA program [53].

| Coal & sustainable biomass | | Natural gas | Electricity |
|---|---|---|---|
| Revamped BF | Greenfield | Revamped DR | Greenfield |
| ULCOS-BF | HIsarna | ULCORED | ULCOWIN ULCOLYSIS |
| tuyere tests in Dunkerque | demonstrator under experimentation | reformulated as HYBRIT | scale-up as SIDERWIN |

**Figure 21.** The ULCOS solutions and their present status.

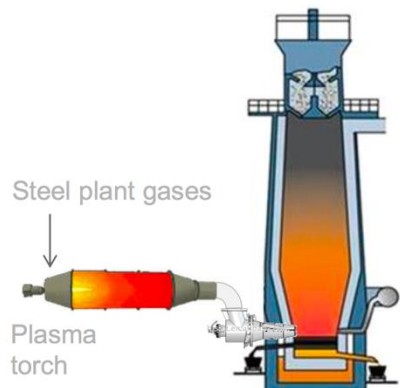

**Figure 22.** The explained I terms of what the acronym means project (source: ArcelorMittal).

The Smelting Reduction *HISARNA* process is continuing at a demonstrator scale in the Tata Steel mill of IJmuiden, Netherlands, with continuing financial support of the EC, through the RFCS and H2020 programs. It is presently conducting long-duration (weeks) trials.

The *ULCORED* process is now continuing as the HYBRIT project, thus as a synthesis of two ULCOS subprojects.

The electrolysis solutions, *ULCOWIN* and *ULCOLYSIS*, are presently continuing as the *SIDERWIN* project, a H2020 project [54,55], and as part of the VALERCO project, respectively, after various EU and National support schemes (EC's RFCS IERO, French ANR's ASCOPE, etc.), cf. Figure 23. SIDERWIN is presently designing a TRL 6 demonstrator that will be able to produce 100 kg of iron.

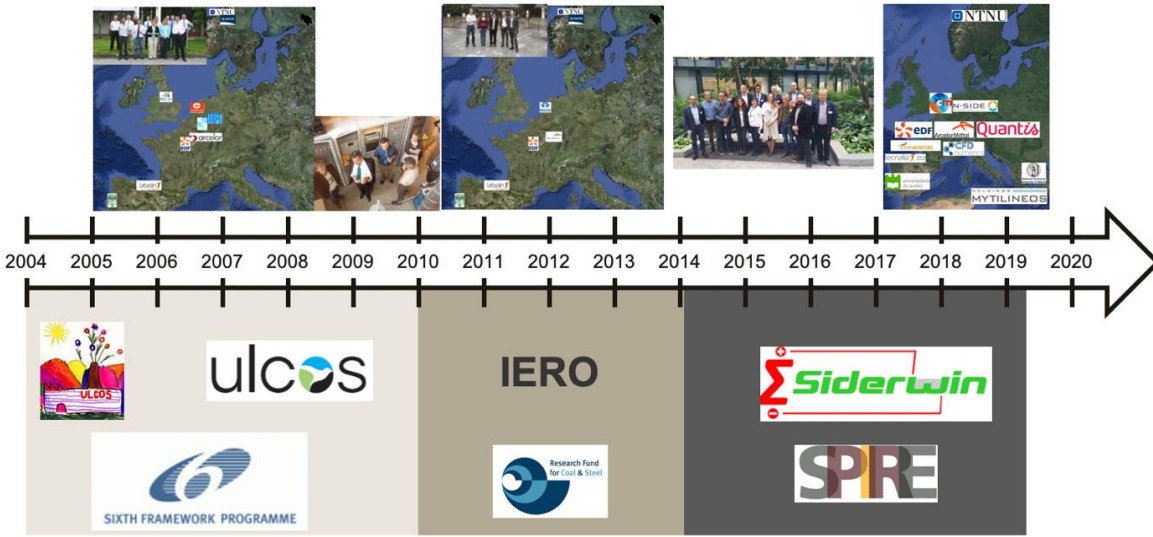

**Figure 23.** Evolution of the low-temperature electrolysis of iron ore, known as ULCOLYSIS and now as SIDERWIN (courtesy of ArcelorMittal).

A 48-month project called 3D-DMX (DMX Demonstration Dunkirk; DMX is the name of the amine-based capture technology) was launched in Dunkirk in May 2019 around ArcelorMittal's steel mill with EU financial support [56] in order to continue the development of a new technology for $CO_2$ capture carried out within the VALORCO project at IFP Energies Nouvelles's laboratory in Solaize, and demonstrate it on a stream of blast furnace gas (BFG) at the scale of the capture of 0.5 $t_{CO2}$/h [57,58]. The next step would be a large-scale demonstrator of 1 Mt/year and there are plans to gather a cluster of emitters around the North Sea and to use existing infrastructure for transport and storage, and thus to connect to the *Northern Lights project*, also funded under H2020 [59] and organized around the Sleipner Platform, where Statoil, now Equinor, developed know-how on the capture and storage of $CO_2$ extracted from the local natural gas stream.

Another new project is related to the decision of ArcelorMittal in Hamburg to build a demonstrator of *hydrogen-based reduction* with Midrex, based on the latter's technology of direct reduction [60]. The target is the production of 100,000 t/year production of direct reduced pellets. Larger scale-up would be envisaged depending on the technical outcome and political context. The other DRI technology providers are also working on their own concept of an $H_2$-based reduction furnace, for example the ENERGIRON ZR process [61].

The efforts carried out in Europe in terms of low-carbon steelmaking—now coined "zero-carbon steelmaking"—reflect the European Commission's commitment to transform Europe into the world's first climate-neutral continent by 2050, a policy known as the European Green Deal [62].

## 5.5. Low-Carbon Steelmaking Solutions in the US

In the US, two projects are aimed at low-carbon technology for steelmaking.

One is the Molten Oxide Electrolysis (MOE) project, a concept proposed by Donald Sadoway at MIT, which is in principle very similar to the ULCOLYSIS process, i.e., a transposition to iron of the Hall–Héroult process of making aluminum, without its drawbacks. The process takes place at liquid metal temperature (≥1538 °C) and metal and iron ore are both molten (cf. Figure 24). MIT has launched a spinoff company, Boston Metals, to develop the technology at a larger scale [63]. The challenge is to find an anode that is not made of carbon, which is what happens in aluminum cells, where it participates in the reduction by evolving $CO_2$ rather than simply $O_2$.

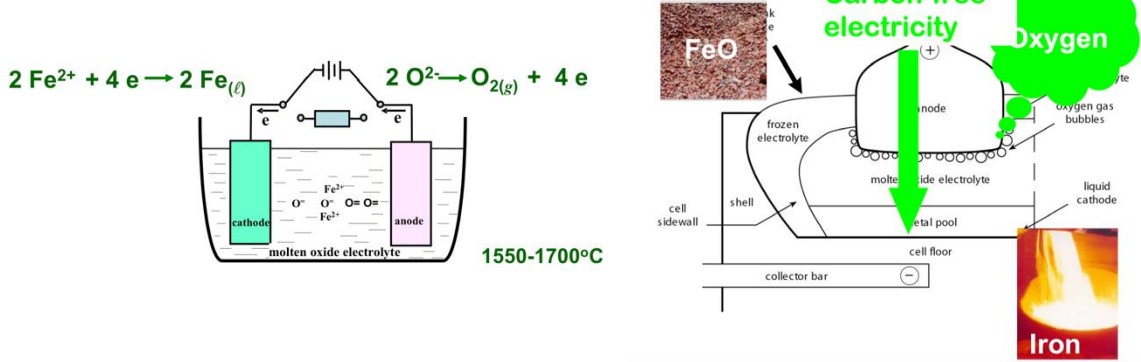

**Figure 24.** The MOE process of D. Sadoway.

The second process originates from the University of Utah (Professor Hong Yong Sohn) and is somewhat similar to the SUSTEEL project already mentioned, except that the reactor concept here is a *flash smelter*, thus operating at very high temperatures in flight on particulate hematite, cf. Figure 25. It is called the new flash smelting process or the flash ironmaking technology (FIT). It receives support from AISI and US' DOE. The project operates a mini-pilot reactor, capable of 1550–1700 °C temperatures with a concentrate feeding rate of 2–5 kg/h and there are plans to scale it up [64].

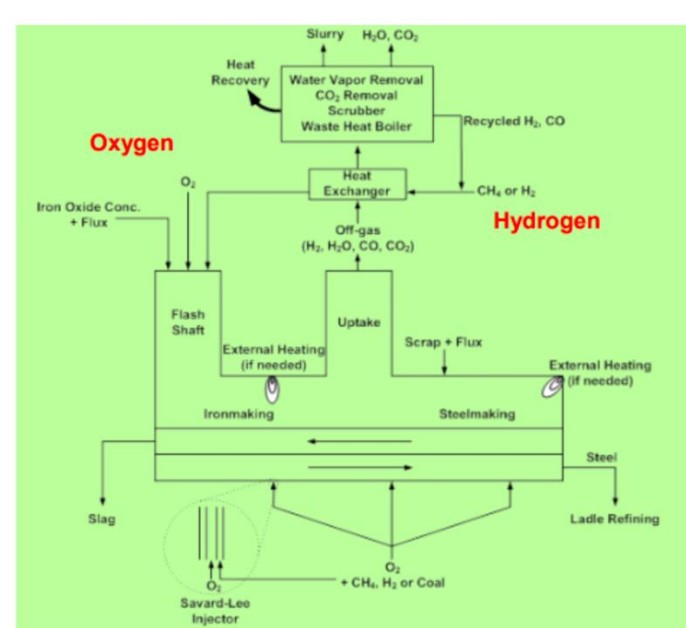

**Figure 25.** New flash melting process, University of Utah.

### 5.6. Low-Carbon Steelmaking Solutions in Japan

Japan has a national project, called COURSE 50 ($CO_2$ Ultimate Reduction in Steelmaking process by innovative technology for cool Earth 50) supported by NEDO, launched in 2007, and presently continuing in its third installment [65]. It collects a series of technology improvements and breakthroughs, the sum of which would add up to a significant reduction in emissions (30% by 2050), cf. Figure 26. It comprises work on CCS but has also a focus on hydrogen, cf. Figure 27.

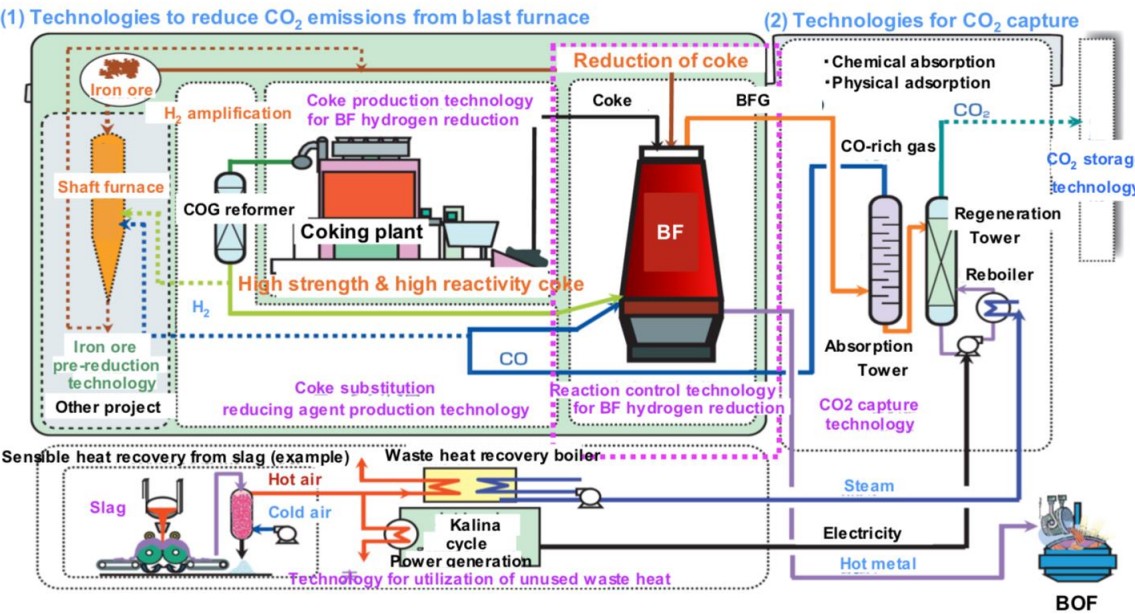

**Figure 26.** The Japanese COURSE50 program (source: ISIJ).

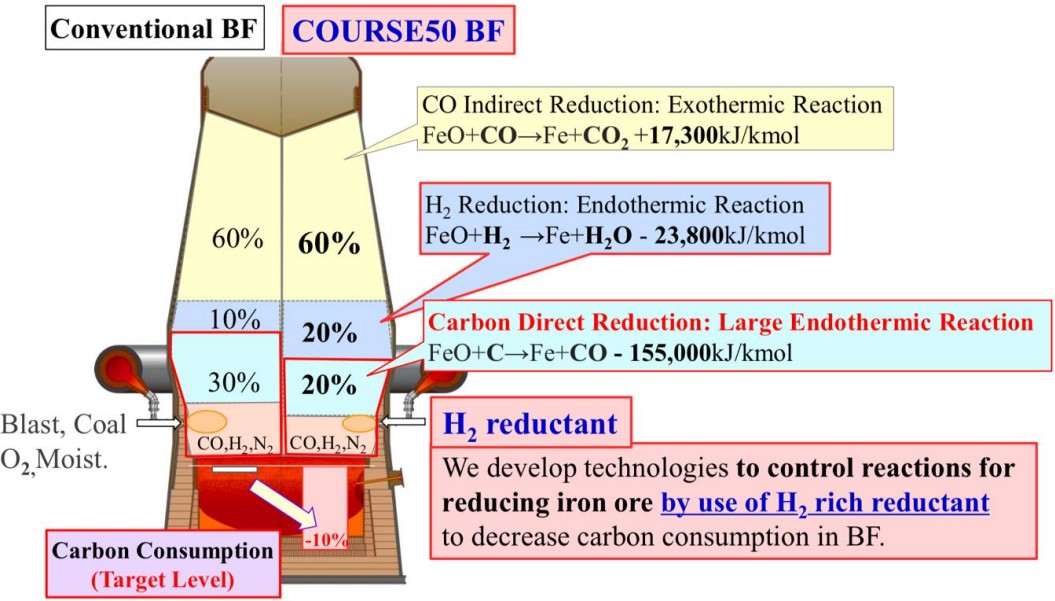

**Figure 27.** Concept of "iron ore hydrogen reduction reinforcement" in the Japanese COURSE50 program (source: ISIJ).

The idea of the "iron ore hydrogen reduction reinforcement" is to use hydrogen inside an otherwise classical BF by injecting a hydrogen-rich mixture at the tuyeres and then replacing the cohesive zone by two regions: one at the bottom where direct reduction of iron ore by carbon takes place and, on top of it, a zone in which hydrogen carries out the reduction. As both reactions are endothermic, energy is brought from the combustion at the nose of the tuyeres and from the descending burden where CO indirect reduction, which is exothermic, has been taking place. It is expected to reduce carbon consumption and therefore $CO_2$ emissions by 10% and a BF demonstrator, similar to the LKAB experimental furnace in Luleå, was built by NEDO to test the concept. Practical implementation is scheduled for beyond 2030.

To bridge the gap towards zero-emissions beyond the COURSE 50 program, Japan has developed a vision to reach carbon neutrality in the steel sector by 2100 [66]. It is based on the development

and the implementation of a series of "super-innovative technologies", pushing COURSE50 targets towards more utilization of hydrogen in the blast furnace (super COURSE50 program), hydrogen reduction, and the systematic use of CCUS. The timeline is shown in Figure 28.

| Development of technologies specific to iron & steel sector | | 2020 | 2030 | 2040 | 2050 | 2100 |
|---|---|---|---|---|---|---|
| COURSE50 | Raising ratio of H2 reduction in blast furnace using internal H2 (COG) Capturing CO2 from blast furnace gas for storage | R&D | | Implementation | | |
| Super COURSE50 | Further H2 reduction in blast furnace by adding H2 from outside (assuming massive carbon-free H2 supply becomes available) | Stepping up R&D | | | | |
| H2 reduction iron making | H2 reduction iron making without using coal | Stepping up | R&D | Implementation | | |
| CCS | Recovery of CO2 from byproduct gases. | R&D | Implementation | | | |
| CCU | Carbon recycling from byproduct gases | | R&D | Implementation | | |

| Development of common fundamental technologies for society | | 2020 | 2030 | 2040 | 2050 | 2100 |
|---|---|---|---|---|---|---|
| Carbon-free Power | Carbon-free power sources (nuclear, renewables, fossil+CCS) Advanced transmission, power storage, etc. | R&D | Implementation | | | |
| Carbon-free H2 | Technical development of low cost and massive amount of hydrogen production, transfer and storage | R&D | Implementation | | | |
| CCS/CCU | Technical development on CO2 capture and strage/usage Solving social issues (location, PA, etc.) | R&D | Implementation | | | |

**Figure 28.** Long-term vision of the Japanese iron and steel industry for zero-carbon steelmaking [66].

### 5.7. Low-Carbon Steelmaking Solutions in China and Other Parts of the World

China, which is the largest steel producer in the world and therefore its prime sectoral $CO_2$ emitter, has been involved in low-carbon technologies since the onset of the $CO_2$ Breakthrough Program (CBP), cf. Section 5.9.

It communicates mostly on energy saving measures, CCU, and emphasizes the "social value" of steel, which helps other sectors and society in general control their emissions. Note that a CCUS center was created in Guangdong in cooperation with the UK and Australia, cf. Figure 29 [67].

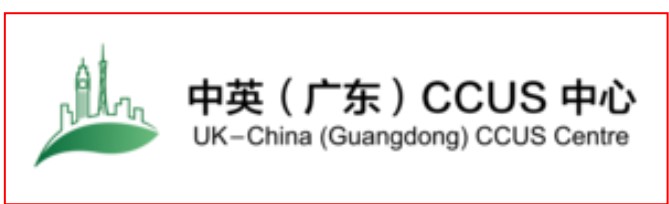

**Figure 29.** The Canton CCUS center's logo.

India, especially Tata Steel, focuses on its energy efficiency accomplishments and the HISARNA demonstrator project running in its IJmuiden steel mill.

Australia is also involved in low-carbon technologies, especially focusing on the use of biomass, as a source of "green" carbon [68].

### 5.8. Carbon Capture and Usage (CCU)

Due to the alleged rejection of CCS by citizens in some parts of Europe, the idea of using $CO_2$ has been pushed forward, mainly by the chemical industry, which is wary of the announced peak oil and peak gas. Today, $CO_2$ is indeed used in the synthesis of urea and a broad array of reaction paths were explored to transform $CO_2$ into virtually any inorganic molecule, including fuels. Estimates of 5% to 15% of potential applications of CCU in terms of available industrial $CO_2$ has been put forward. Note that innovative uses of CO are also studied.

In Europe, the concept is being eagerly explored in various projects, including the VALERCO project already mentioned, but also in more recent ones, which have received public financial support:

- Carbon2Chem, proposed by ThyssenKrupp steel, Germany [69]. The concept is to make various chemicals, including ammonia, from steel mill gases (44% nitrogen, 23% carbon monoxide, 21% carbon dioxide, 10% hydrogen, and 2% methane) and with the application of renewable energy, which has the reputation of being carbon neutral. The project has support from the Federal Ministry of Education and Research, Germany. Cf. Figure 30.
- Carbon4PUR, involving ArcelorMittal and Dechema et al., is a SPIRE/H2020 Project [70]. The focus is on the production of polyurethane (PUR) foam and coatings from steel mill gases. Cf. Figure 31.
- STEELANOL, involving ArcelorMittal, Belgium (ethanol) and using Lanzatech technology, targets the transformation of CO from converter gas (BOG) by using fermentation driven by bacteria [71]. Cf. Figure 32.
- Other European projects focusing on CCU are FReSMe, ICO2CHEM, MefCO2, and RECO2DE [72].

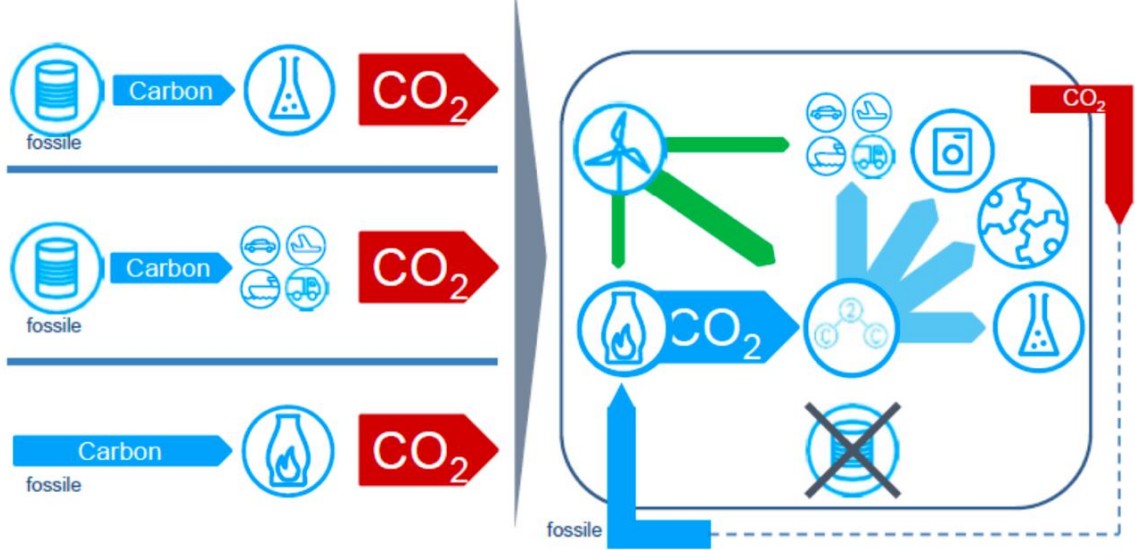

**Figure 30.** The Carbon2Chem project (source: ThyssenKrupp Steel).

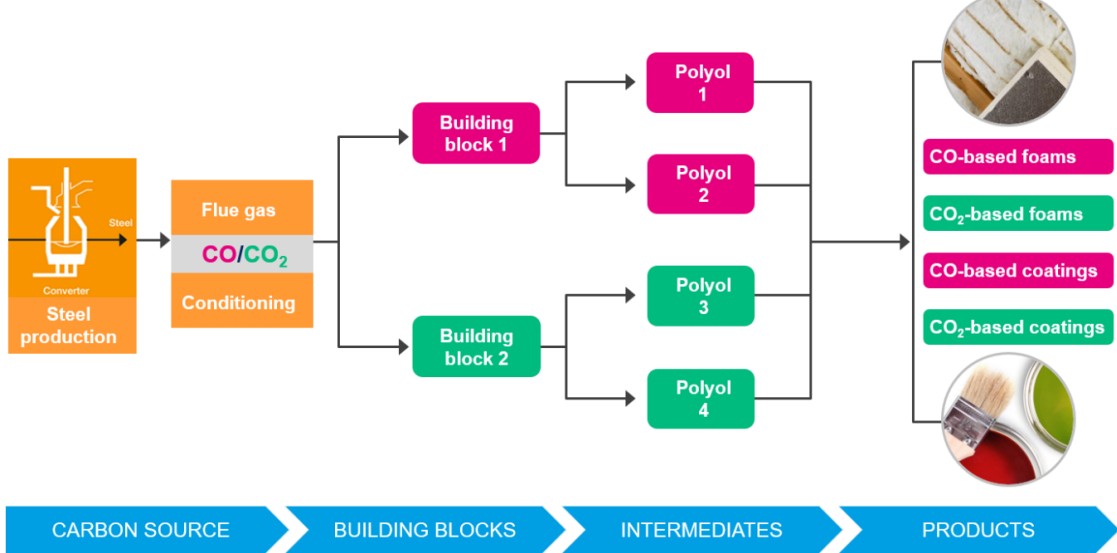

**Figure 31.** The Carbon4Pur project concept (source: ThyssenKrupp Steel).

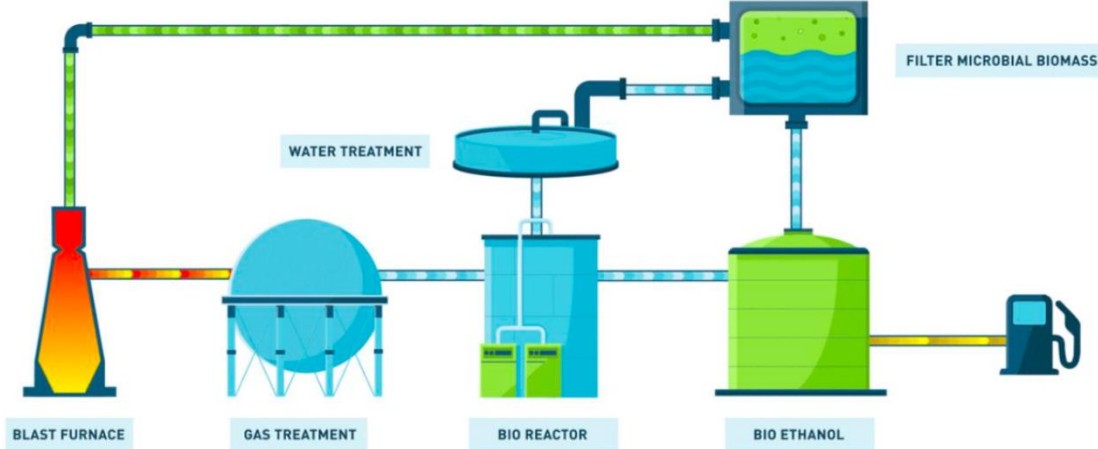

**Figure 32.** The STEELANOL project (source: ArcelorMittal).

These projects are presented as offering an opportunity for several energy-intensive industries to work together and, implicitly, to arrive at more efficient solutions than by acting as a single sector. Moreover, the claims that CCU reduces GHG emissions has to be substantiated by carrying out consequential LCA studies: is it indeed cutting emissions or simply postponing them by a few weeks or months? Does it avoid the use of fossil fuel or simply leaves it available for some other usage?

*5.9. Institutional Support to Low-Carbon Steelmaking Technologies*

Various institutions and organizations have been supporting the development of new low-carbon steelmaking technologies. The rationale is, on the one hand, that only breakthrough solutions could deliver the very low emissions that climate change requires, and, on the other hand, that this would call for a tremendous effort in terms of time, continuity of purpose, and financial support.

While the initial work on the topic goes back to the late 1980s and the early 1990s [73], the steel sector started to organize in the early 2000s and lobbied for financial support in the EU and in various countries from national funding agencies from that time on. This led to the *ULCOS program*, supported by the European Commission under the RFCS and Framework programs, and to the *$CO_2$ Breakthrough Program* (CBP) of Worldsteel, where participants from all over the world would seek support from their own countries. For a period of about 10 years, research on low-carbon steelmaking was conducted under a single banner, with programs in the 5–50 M€ range or slightly more.

ULCOS was about to launch a second stage, ULCOS II, with a budget size an order of magnitude larger than the previous generation of projects. Financing had been arranged through the *NER 300 program* until the ULCOS-BF project collapsed due to the shutdown of the Florange Blast Furnace, decided as a consequence of the 2008 economic crisis.

Since then, the various projects have been continuing individually and new directions are presently being explored (mainly hydrogen processes and carbon utilization). A so-called "big ticket" program is in the making in Europe, under the auspices of EUROFER and ESTEP and with preliminary support from the EC to an RFCS project (*LowCarbonFuture*) that should define its scope. The future ETS Innovation Fund, also called NER 400, is involved [73]. Work is continuing also in Japan, with a large size program (*COURSE50*) and the project to extend it further.

Institutions like the *IEA* and the *Steel Committee of OECD* have picked up an interest in the topic and the IEA is presently working on a *sectoral roadmap for steel* [74].

*5.10. Production of Hydrogen*

It is often taken for granted, in recent literature about hydrogen reduction, that hydrogen will be produced from electricity by *electrolysis of water* or, in the future, of steam (high-temperature electrolysis).

This would go along with a huge increase in electricity production capacity: a 5 Mt/year steel mill would require a 1200 MW nuclear power plant or 240 recent wind turbines. The electrolysis technology is not yet ready for immediate scale up to this large size, but R&D is ongoing.

One should therefore take on board the results shown in Figures 18–20, which indicate that the traditional technology based on the *steam reforming of methane* can continue to be used, especially if some technology improvements are introduced, such as CCS or the production of the less pure hydrogen that the steel sector is likely to need. The use of the hydrogen present in COG would also be of interest, cf. Section 5.1.

*5.11. Conclusions on Zero-Carbon Steelmaking*

There is no doubt that steel will continue to be produced *en masse* in the 21st century, and that production will keep increasing to accommodate the accession of more people to a decent standard of living across the world. Therefore, ways to alleviate the GHG emissions of the sector will need to be implemented and they will involve major changes in the way of making steel, thus they will need to be based on "breakthrough technologies".

There are myriad solutions for carrying out low-carbon steelmaking.

The future of *hydrogen reduction* depends on the future of hydrogen in general. The *hydrogen society* was brokered as a dream by people like Jeremy Rifkin [75], but the dream may be easier to implement in outer space than on Earth! Various sectors will compete for hydrogen, for example, transport and steel and transport today would be ready to pay more for hydrogen than steel. The value-in-use in each sector is quite different, an order of magnitude at least!

Moreover, if hydrogen use does not take off for transport, i.e., if fuel cell electric vehicles do not compete seriously with electric battery vehicles, then it is unlikely that it will be used in the steel sector.

Additionally, hydrogen reduction will compete with the direct electrolysis of iron ore. This process is developing briskly, and it could seriously be a credible alternative to "green" hydrogen reduction in 2030 and beyond. On paper, it looks like it would be more energy efficient than green hydrogen reduction, but this needs experimental confirmation to be completely believable.

Additionally, the continuing use of coal and natural gas, associated with CCS or CCU will probably remain large in the steel sector.

Thus, research and development have to continue, in parallel, exploring the many possible low-carbon routes to steelmaking.

## 6. Steel, Biodiversity, and Ecosystem Damages

The loss of biodiversity has today reached a crisis level, becoming the sixth major extinction since the advent of life on earth [76]. This is certainly one of the major threats to "the environment", caused by a combination of factors, including climate change and the growing urbanization of the world. Biodiversity loss in addition to the loss of million species also means the loss of ecosystem services that biodiversity usually brings. Any human activity, individual, collective or industrial has a responsibility in this erosion of biodiversity, as it influences its causes. However, the connection with steel and steel production is not specific and therefore will not be discussed further here [77].

## 7. Steel, Health of People, Animals and Ecosystems

In biological matter (*biochemistry*), iron is an essential element in the fabric of life [78]. Steel, i.e., iron and its alloying elements, may also act as a toxicant, and as such is studied by *toxicology*. Steel production also raises health issues in the workplace (*occupational health*) or around steel mills (*public health*): both subjects are covered by *epidemiology*. The connection between steel mills and people's contamination is an environmental issue related to the phenomena of *emissions and pollution*. For example, the creation of an environmental department at IRSID in the 1970s was related to an industrial accident, the intoxication of cattle near the steel mill of Le Breuil in Le Creusot in France, by molybdenum emissions from the electric arc furnace. Therefore, there has been a strong connection between emissions, pollution, and "public" health, since the early beginnings of the discipline. Beyond effects on human health, the environment also affects life in general and, more broadly, ecosystems, studied by specific disciplines, *ecotoxicology* and *ecoepidemiology*. Moreover, the field, globally, is covered by an emerging discipline called *environmental health*. It is defined by WHO in its 1989 conference in Frankfurt as being "related to aspects of human health and diseases, which are driven by the environment. This also refers to the theory and practice of controlling and measuring environmental factors that may potentially affect health" [79].

The role of iron in the biochemistry of animals, from microorganisms to human beings, is related to the toggle between the two redox states, $Fe^{2+}$ and $Fe^{3+}$, a mechanism used to transfer electrons inside cells and thus participate to its *metabolism*. Important enzymes (ferritin, transferrin), the hemoglobin of blood cells (aptly called *hematies*), and other biomolecules contain iron. Fe is therefore an essential trace element in the human diet, with health issues if there is too much or too little present in the daily input (7–11 g/day) [78].

This, however, can be considered as unrelated to the iron and steel sector, because *the geobiochemical cycle of iron* is mainly disconnected from the *anthropogenic iron cycle* [80].

*Toxic metals*, usually referred to as *heavy metals*, include the following [81]:

- Plant toxicity: Al, As, Cd, Cr, Cu, Pb, Mn, Hg, Ni, Pt, Se, Ag, Th, W, U, V, Zn. Animal toxicity: Al, As, Cd, Cr, Cu, Pb, Li, Mn, Hg, Se, Th, Sn, W, U, V, Zn.

While the following metals are essential to life (thus, some are both toxic and essential at different levels):

- Essential to plants: Cu, Fe, Mg, Mn, V.
- Essential to animals: Cr, Co, Cu, Fe, Mg, Mn, Ni, Se, Sn, V, Zn.

*Iron* is both *essential* and *non-toxic* to any of the life kingdoms, but many of the elements used in alloys do raise health concerns (Al, Cr, Cu, Pb, Mn, Ni, Se, W, Sn, Zn). Note that animal and human toxicity are not identical and that there are many levels of toxicity (acute vs. chronic, lethal or not, confirmed vs. alleged, etc.); see a specialized source for details [83]. Moreover, there are important ongoing controversies regarding the toxicity of common metals like aluminum, chromium, or nickel. Lastly, other authors have published slightly different lists of toxicant metals (for example, Mo is not in the lists). In a practical way, toxic problems are most acute in the working environment of steel mills and, therefore, are handled as part of *occupational health and safety*, mostly satisfactorily nowadays in world-class steel mills [84].

Regarding air emissions caused by the production of steel (cf. Section 4), there are various factors that raise concern:

- Particulate matter or dust: TPM, $PM_{10}$, and now $PM_{2.5}$ are routinely measured nowadays [85], but critics point out the fact that smaller particles than 2.5 µm (or 1 for PM 1) and particularly nanoparticles are still unaccounted for, even though they are very likely to penetrate deeply into the body of living organisms and, thus, to be raising the most serious health risks;
- PAHs;

- PCBs;
- PCDDs and PCDFs;
- HCB;
- More generally, VOCs;
- POPs; and
- Inorganic pollutants ($SO_2$, $H_2S$, $NH_3$, HCl, HCN, $HNO_3$, $O_3$, CO, $CO_2$, black carbon, etc.) and products of incomplete combustion (NOx). Note that inorganic and organic pollutants can combine, such as dust particles made of a core of black carbon and a coating of PAHs.

Steel processes are direct sources of all of these emissions and the standard way to control them is to capture the fumes and "treat" them with a technology that is deemed to deliver the level of purification required by regulations. Discussion of the best *abatement technologies* lies outside of the scope of this paper [20]. Direct process emissions have the reputation of being properly captured. Emissions that escape to the atmosphere (like canopy emissions or other uncaptured emissions) are dispersed in the environment. Sometimes, the issue disappears because of *dilution*, but, in other cases *uncaptured emissions* aggregate with emissions from other sources and contribute to *large-scale pollution* and to their associated health problems.

Most uncaptured emissions are related to particulate matter, cf. Section 4.2. Figure 33 shows the evolution of $PM_{10}$ emissions in EEA-33 Europe [86]. Industry-related emissions (green and beige), part of which come from the steel sector, account for more than 1/3 of total emissions and their absolute amount has only slightly decreased over the past 30 years. The data for $PM_{2.5}$ are not available for this extended time period. In 2017, regarding $PM_{10}$ and $PM_{2.5}$, industry was responsible, respectively, for 17% and 10% of the total emissions [87]. The EEA reports on the largest polluters in Europe. In 2015, 7 of the 10 top polluters in $PM_{10}$ were integrated steel mills [88].

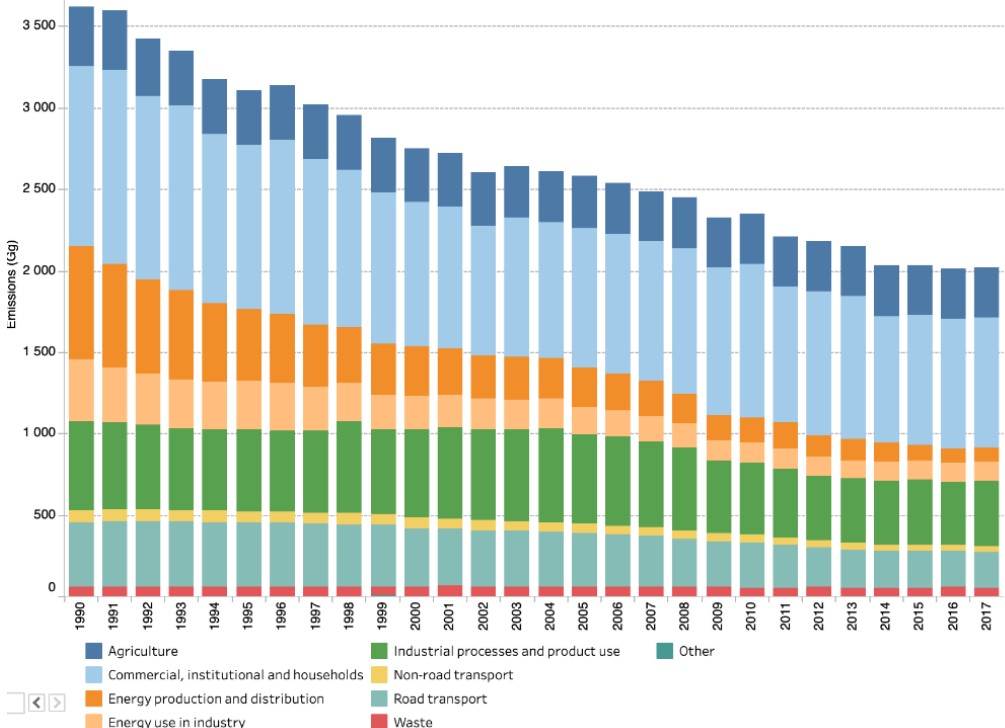

**Figure 33.** $PM_{10}$ emissions by economic sector in Europe (EU-28). Source: European Environment Agency [82].

Air pollution has serious health effects:

- WHO reports 4.2 million premature deaths ("death that occurs before the average age of death in a certain population", according to the NIH) due to outdoor pollution (worldwide ambient air pollution accounts for 29% of all deaths and disease from lung cancer, 17% from acute lower respiratory infection, 24% from stroke, 25% from ischemic heart disease, 43% from chronic obstructive pulmonary disease. Pollutants with the strongest evidence for public health concern, include particulate matter (PM), ozone ($O_3$), nitrogen dioxide ($NO_2$), and sulphur dioxide ($SO_2$)).
- 3.8 million premature deaths are due to indoor pollution worldwide, as shown in Figure 34 [89].

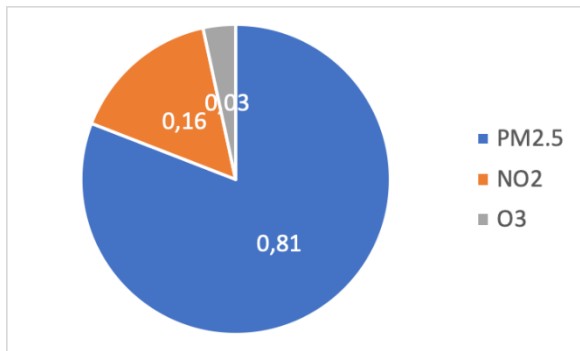

**Figure 34.** Repartition of air pollution in terms of premature deaths due to $PM_{2.5}$, $NO_2$, and $O_3$. for EU-28 in 2015.

Premature deaths were essentially due to $PM_{2.5}$, $NO_2$, and $O_3$, in the proportions of 81%, 16%, and 3% in EU-28 in 2015, i.e., 483,400 deaths, of which 391,000 are attributed to microscopic particulate matter [90].

Allocating part of these premature deaths to the steel sector is difficult due to a lack of fine-grained data and of relevant literature. More generally, this matter of premature deaths as related to air pollution should be examined at the scale of industrial sectors or activities like transport, whereas WHO has been carrying it out at an aggregated world level.

There is, however, a troth of literature devoted to emissions from steel mills, people's impregnation by pollutants, and health. However, even though all contribute some relevant information, none establish causal relationships between emissions and health, beyond undocumented statements, militant guesses, rumors, and even plain fake news. The difficulty is that a steel mill is usually part of an industrial complex, which generates a variety of emissions that aggregate to cause local pollution. This was analyzed in detail, for example, in the case of the Vitória metropolitan area (cf. Section 4.5). The literature comprises reports of militant organizations (NGOs) [91], their counterpart from steel business, i.e., CSR reports [92], special reports on local case studies in Taranto, Italy [93,94] and Fos-sur-Mer, France [95,96], and more.

In the future, when all society stakeholders will continue to seek reduced emissions of particulate matter, the steel sector will have to participate in that effort [97]. This will call on new investment and, probably, on the development of new measuring devices and new abatement technologies.

Reviews will be needed, but probably also new studies, to clarify whatever data and knowledge are available today.

## 8. Conclusions

This review paper has looked at steel, and particularly steel production, as it is related to the whole spectrum of environmental issues. This was done in two steps.

The first step consisted in taking stock of the fact that the main issues have been properly studied, understood, and kept under control:

- Iron resources are abundant, both primary and secondary raw materials: indeed, iron ore resources and reserves are plentiful, and steel is fully active in the circular economy, being the most recycled material.

- Energy consumption is already lean, an evolution driven by the high energy intensity of the sector and the high prices of energy. Change will take place by incorporating more renewables in the sector's energy mix, implementing the energy transition in an original way (electrification, CCUS, and/or use of green hydrogen), and codeveloping lean, frugal, and more durable product solutions with steel users, as well as switching to PSSs.

- Emissions to air, water, and soil have been curbed and most of them have started to decrease, at least in countries with advanced state-of-the-art steel mills. Among all air emissions, rogue emissions of particulate matter (coarse $PM_{10}$, fine $PM_{2.5}$, and ultrafine nanoparticles) probably still need to be better measured and better captured than it is yet the case today.

- We did not linger on the looming major biodiversity extinction, because the connection of this phenomenon with a particular metal like steel is only indirect, through its contribution to climate change and to urbanization of the anthroposphere.

- Steel is responsible for a sizeable chunk of greenhouse gas emissions, generating roughly twice as much $CO_2$ as the quantity of steel produced. However, the sector started to explore radical solutions for cutting emissions early, and a large number of them have been identified and tested at some intermediary scale, laboratory, pilot, or demonstrator.

The second step consisted in exploring the remaining open and unsolved issues:

- Regarding raw materials, mine tailings are still mostly out of control, with too many dams failing regularly across the world, thus creating major industrial disasters. All metals bear a similar responsibility.

- Emissions, particularly air emissions, still constitute a major problem in and around steel mills. The main remaining issue is particulate matter, which is generated by most human activities but also and significantly by steel: out of the 10 largest polluters in Europe, 7 are steel mills! The key reason for worrying about PM is the number of premature deaths that it causes.

- The third major task is to arrive at practical solutions to reach carbon neutrality by 2050, a commitment that all stakeholders in Europe have made and that they have to materialize.

Today, in the first quarter of the 21st century, the world has abandoned the irenic view that progress can easily improve the standard of living of most people on Earth and, at the same time, preserve nature and leave the environment intact for future generations. In the case of steel, this means aggressively addressing the major issues that are still open, and finding practical and working process solutions. This will mean radically redesigning steel production with even more demanding environmental targets in mind.

It is no longer possible today to treat environmental issues at the margin, like in an extra chapter of a metallurgy treatise. They need to be taken on board at the onset of any major technical action.

**Funding:** This research received no external funding.

**Conflicts of Interest:** The author declares no conflict of interest.

## Glossary

| | |
|---|---|
| AP | acidification potential |
| BC | Black Carbon |
| CCS | Carbon capture and storage |
| CCU | Carbon capture and usage |
| CCUS | Carbon capture, usage and storage |
| COURSE 50 | $CO_2$ Ultimate Reduction in Steelmaking process by innovative technology for cool Earth 50 (Japan) |
| EAF | Electric Arc Furnace |
| EEA | European Environment Agency (EU) |
| EP | eutrophication potential |
| GHG | Greenhouse gases |
| GWP | Global Warming potential |
| HCB | Hexachlorobenzene |
| HM | Heavy Metal |
| HRC | Hot-rolled coil |
| IRSID | Institut de recherches de la sidérurgie française (France) |
| IPBES | Intergovernmental Science-Policy Platform on Biodiversity and Ecosystem Services (UN) |
| ISF | Imperial Smelting Furnace (zinc blast furnace) |
| LCA | Life Cycle Assessment or Life Cycle Analysis |
| MIT | Massachusetts Institute of Technology (US) |
| Mg | megagram = ton |
| NEDO | New Energy and Industrial Technology Development Organization (Japan) |
| NIH | National Cancer Institute (US) |
| NMVOC | Non-methane volatile organic compound |
| PAHs | Poly-Aromatic Hydrocarbon |
| PIA | Programme d'investissements d'avenir (France) |
| PCB | Polychlorinated biphenyl |
| PCDD | polychlorodibenzodioxin |
| PCDF | polychlorodibenzofuran |
| PCDF | polychlorodibenzofuran |
| PED | Primary energy demand |
| PM | Particulate Matter |
| $PM_{10}$ | Particulate Matter with dimensions less than 10 μm |
| $PM_{2.5}$ | Particulate Matter with dimensions less than 2.5 μm |
| POCP | photochemical ozone creation |
| POP | persistent organic pollutant |
| PSS | Product Service System |
| PUR | polyurethane |
| SGPI | Secrétariat général pour l'investissement (France) |
| TSP | Total Suspended Particles (<100 μm) |
| ULCOS | Ultra-LOw $CO_2$ Steelmaking |
| USGS | United States Geological Survey |
| VOC | Volatile Organic Compound |
| WHO | World Health Organization, a specialized agency of the UN (*OMS* in French) |

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
