# Peer review of "Society, Materials, and the Environment: The Case of Steel"

_metals, doi:10.3390/met10030331_

Round 1

Reviewer 1 Report

The author is to be congratulated for addressing what is probably the most important and difficult topic currently facing the world steel industry - the impact of steel production and use on environmental issues. The review is very wide in scope and aims (production, emissions, technology breakthroughs, lifecycle, usage, biodiversity damage, health issues…) but in reality only the first four subjects are treated in any detail. The text contains a wealth of useful information and analysis. However, it appears to have been assembled from multiple sources and reports as evidenced by some rather abrupt changes in font, formatting and section numbering. Readability could be greatly improved if a more uniform style could be adopted. Also, many of the figures look more suitable for powerpoint presentations than a journal article – for the most complex diagrams more description in the text would be appreciated.

The author might want to consider removing sections 6 and 7 to make the remaining text more focussed.

Suggested corrections:

There is a problem with the section numbering as the section headings are all 1 digit in advance of the sub-headings. This makes references to other sections tricky to follow.

A second sub-section 1.1 is introduced on page 25, line 642.

Numbers inserted in the text need to be checked e.g. lines 126,127,134, 566, 567.

Line 97: economy

Line 201: meek?

Line 229: caused

Line 247: GHG

Line 353: palletisation

Line 358 – line is incomplete

Line 421: condense

Line 463: needs rewriting

Line 583: impersonation?

Line 605: latter

Line 690: needs a reference

Line 743: Should read Figure 28?

Line 798: carbon

Line 966: premature deaths

Line 1014: cloud

Figure 11: reference not defined

Figure 19 and Figure 20 – same caption?

References 11, 73, 85, 90, 92, 93, 95 do not appear to be complete?

Author Response

Thank you for such a detailed and careful reviewing of my paper.

I tried to answer each of your questions.

I enclose a file that contains your questions and my answers.

Reviewer 2 Report

Abstract and keywords: The journal guideline indicates 3 to 10 keywords and the paper has 24. Please avoid using so much terms and summarize only pertinent keywords.

The abstract is too long – more than 700 words.

Use the same font in the entire document.

“The present article discusses the connection between the environment and steel, both production and use.” What about end-of-life? What is the contribution of this review? Is it intended to answer something? Fill some gap?

In every section the author initiates with: “Issues discussed in section 1: raw materials, location/concentration/resources & reserves of iron ore mines, […]”. I suggest removing it.

Line 134: 20007 = 2007.

Figure 1 and 2– include axis labels and the source.

Figure 3 – This figure was founded in the following link: http://www.groundtruthtrekking.org/Graphics/Mining-terms.html
Please include the source.

Lines 184 – 186 – Why this text is in italic?

Lines 187 – 192 – It is important to highlight that the types of system boundaries in LCA studies usually are: Cradle-to-gate, Cradle-to-grave and Gate-to-Gate.  Please explain more how LCA could include an indicator for dam failure.  In my opinion this is an issue to be dealt in risk analysis and which falls outside the scope of the possible applications of an LCA.

“Furthermore, the full upstream mass balance related to raw materials may or may not be included, depending on […] and on whether the overburden is overlooked or not.” Please further explain this.

Figure 5 should be better explained and discussed in the text and most importantly, should include the source of it (VDEh blast furnace comitee).

Lines 269-275 – please format the text (same font and size).

Lines 298-299 – please explain more about the relationship of PSS + steel + reduce energy consumption, especially the example of a short-term rental scheme apartment.

Lines 324-328:  

“Emissions are analyzed at a steelmill's or a single reactor's scale, either as statistical data on
emissions or as process engineering analyses of how the emissions are generated.”

Do you mean in general literature?  

“The global phenomenon of atmospheric pollution is only analyzed by meteorological tools.”

Modeling of dispersion of air pollutants is an important and challenging scientific problem that has beend adressed by diffrent scientifc domains according to environmental impacts and effects on human health. Please elaborate more on this affirmation.

“ Emissions of the value chain, especially of its upstream part, should also be part of the discussion.” This is oftleny done in LCA studies.

Figure 9 – Include source.

Lines 440-444 – Reference of this study?

Figure 14 and 15 – please include the source in the legend.

“3.6. Water and soil emissions: Emissions to water and soil ought to be discussed in parallel to the previous discussion focused on air. We leave this for further reviewing work.” Maybe is better to explain this in another section instead of creating one new topic.

Table 3 – please include the source also in the legend.

Figure 21 and 22 - please include the source also in the legend.

Line 695: “In the US, two projects target low-carbon technology for steelmaking”.

Lines 867-868 : “However, the connection with steel and steel production is not specific and therefore will not be discussed further here”. Be careful with this type of information. For example, would you say that steel making has no relation with habitat loss? Also, ecotoxicology issues are discussed in the next section for example.

Finally, the article presents formatting problems, lack of references in the figures and excessive information. The review could be more concise, summarizing the relevant information.

Author Response

Thank you for reviewing my paper in such detail!

I tried to answer all of your questions to the best of my abalities. 

Round 2

Reviewer 2 Report

The relevant changes were made.